Corrected: Author correction

# TRPS1 shapes YAP/TEAD-dependent transcription in breast cancer cells

Dana Elster[1], Marie Tollot[1], Karin Schlegelmilch [2], Alessandro Ori[1], Andreas Rosenwald[3], Erik Sahai [2] & Björn von Eyss [1]

Yes-associated protein (YAP), the downstream transducer of the Hippo pathway, is a key regulator of organ size, differentiation and tumorigenesis. To uncover Hippo-independent YAP regulators, we performed a genome-wide CRISPR screen that identifies the transcriptional repressor protein Trichorhinophalangeal Syndrome 1 (TRPS1) as a potent repressor of YAP-dependent transactivation. We show that TRPS1 globally regulates YAP-dependent transcription by binding to a large set of joint genomic sites, mainly enhancers. TRPS1 represses YAP-dependent function by recruiting a spectrum of corepressor complexes to joint sites. Loss of TRPS1 leads to activation of enhancers due to increased H3K27 acetylation and an altered promoter–enhancer interaction landscape. *TRPS1* is commonly amplified in breast cancer, which suggests that restrained YAP activity favours tumour growth. High TRPS1 activity is associated with decreased YAP activity and leads to decreased frequency of tumour-infiltrating immune cells. Our study uncovers TRPS1 as an epigenetic regulator of YAP activity in breast cancer.

[1] Leibniz Institute on Aging, Fritz Lipmann Institute e.V., Beutenbergstr. 11, 07745 Jena, Germany. [2] Tumour Cell Biology Laboratory, Francis Crick Institute, 1 Midland Road, London NW1 1AT, UK. [3] Institute of Pathology, University of Würzburg, and Comprehensive Cancer Center Mainfranken (CCCMF), Josef-Schneider-Str. 2, 97080 Würzburg, Germany. These authors contributed equally: Dana Elster, Marie Tollot. Correspondence and requests for materials should be addressed to B.v.E. (email: bjoern.voneyss@leibniz-fli.de)

Yes-associated protein (YAP) acts as a transcriptional coactivator protein downstream of the Hippo pathway, a pathway with remarkable capabilities during regeneration and cancer development[1–4]. The Hippo pathway was initially discovered in the fruit fly, where deregulated activity of the YAP orthologue Yorkie leads to strong overgrowth phenotypes[5]. Since then, many groups have shown that YAP acts as a very potent oncogene in several mammalian tissues, such as the murine liver[6,7]. Surprisingly, high YAP activity is commonly associated with a better survival prognosis for colon and breast cancer patients, qualifying YAP rather as a protein with tumour-suppressive functions in this tumour types[3,8]. One mechanistic explanation for YAP's tumour-suppressive role in breast cancer is that deregulated YAP/TAZ activity in breast cancer cells induces an anti-tumourigenic immunosurveillance response, ultimately leading to the eradication of tumour cells[4]. Breast cancer cells consequently need to select for (epi)genetic changes during tumorigenesis to restrain YAP activity.

Biochemically, the Hippo pathway comprises a core kinase cascade, composed of MST1/2 and LATS1/2. Several upstream stimuli are able to initiate this kinase cascade so that MST1/2 kinases activate the downstream LATS1/2 kinases[9]. In turn, LATS1/2 kinases phosphorylate YAP/TAZ, leading to their cytoplasmic sequestration and/or proteasomal degradation[10,11]. In the absence of active Hippo signalling, YAP/TAZ can shuttle to the nucleus, where they act as potent transcriptional activators, mainly for the TEAD transcription factor family (TEAD1–4). Recent chromatin-immunoprecipitation (ChIP)-Sequencing approaches revealed that even though YAP/TAZ and TEAD show binding to some promoters, e.g. the promoter of the well-described target gene *CTGF*, most of their joint sites lie in enhancer regions. From these distal regions, YAP/TAZ in turn activate transcription of interacting promoters, e.g. by promoting RNA polymerase II pause release[12–14]. Since it has emerged that many Hippo-independent pathways play a central role for the regulation of YAP activity[8,15,16], we set out to identify Hippo-independent regulators of YAP activity in an unbiased manner using a genome-wide CRISPR screening approach.

Hereby, we have identified the transcriptional repressor protein Trichorhinophalangeal Syndrome 1 (TRPS1) as a new repressor of YAP/TEAD-dependent transcription in breast cancer cells by specifically recruiting corepressor complexes to YAP/TEAD sites, altering the chromatin structure and changing enhancer–promoter interactions of YAP target genes. Importantly, *TRPS1* is commonly amplified in breast cancer, required for efficient tumour growth in vivo and TRPS1 activity is strongly anti-correlated with YAP activity in human breast cancer patients.

## Results

**A CRISPR screen identifies new regulators of YAP activity**. To identify modulators of YAP's transcriptional activity that act independently of the canonical Hippo pathway, we generated an MCF10A sensor cell line allowing us to monitor exogenous YAP activity on a cell-by-cell basis (Fig. 1a).

For that, we chose the MCF10A cell line, a primary breast cell line, which has been extensively used in studies on Hippo signalling[17]. The sensor cell line contains two functional elements: a doxycycline-inducible Strep-tagged YAP 5SA allele and a turboRFP reporter driven by a small promoter fragment containing TEAD-binding sites of the well-characterized direct YAP target gene *CTGF*. The YAP 5SA allele harbours mutations in all LATS phosphorylation sites rendering this mutant insensitive to the upstream Hippo pathway[11]. Not only did the induction of the Strep-YAP 5SA transgene by addition of doxycycline lead to a potent induction of the direct YAP target genes *CTGF* and *ANKRD1* but also to a strong induction of

turboRFP expression (Fig. 1a–c; Supplementary Fig. 1a). Moreover, depletion of YAP or TEAD1 by siRNAs in the doxycycline-induced sensor cell line strongly decreased the turboRFP signal (Supplementary Fig. 1a). Thus, the turboRFP reporter provided a faithful measure of YAP 5SA activity.

To screen for modulators of YAP 5SA activity, we infected the sensor cell line with a genome-wide lentiviral CRISPR library (GeCKO v2) targeting every single gene in the human genome by six independent sgRNAs[18]. After infection with the library, Strep-YAP 5SA was induced for 48 h by addition of doxycycline and cells were subsequently sorted by flow cytometry. We sorted out two populations based on the RFP signal: a "low" population (1% of cells with the lowest signal) and a "high" population (1% of cells with the highest signal) (Fig. 1d, e). The frequency of the sgRNAs within these two subpopulations was subsequently determined by next-generation sequencing and compared to the unsorted population. As a positive control, we included three independent sgRNAs targeting the Strep-tag of the inducible Strep-YAP 5SA transgene. As expected, these sgRNAs were highly enriched when comparing the "low" population to the unsorted population (Supplementary Fig. 1b). We integrated the score of individual sgRNAs targeting the same gene using the redundant siRNA activity (RSA) algorithm, as described previously[18]. With this analysis, we were able to identify 625 genes that were significantly enriched (RSA *P*-value < 0.01) in the "low" population and 565 genes in the "high" population (Fig. 1f). Noteworthy, we identified YAP, RHOA and TEAD1 among the top hits of the "low" list. This is consistent with our depletion experiments, where siRNAs targeting YAP and TEAD1 decreased turboRFP reporter activity and previous findings that RHOA has the ability to upregulate YAP activity in a LATS-independent manner[19,20]. Here, we focused on the hits from the "high" list, since we were aiming to identify factors that restrain YAP activity during breast cancer. To eliminate potential false-positive hits, we subsequently followed a stringent validation workflow (Fig. 1g). First, we performed RNA-sequencing of the ethanol (EtOH)-induced sensor cell line and eliminated hits with an RPKM value <1 as they were considered not to be expressed in this cell line and should consequently not score in a loss-of-function screen. Second, we used a small siRNA library targeting the best 39 hits (according to RSA score) from the "high" list, and tested the siRNA's ability to super-activate the RFP reporter under doxycycline-induced conditions. To eliminate those hits that might solely act on the reporter but not on YAP target gene expression per se, we tested the effect of siRNA-mediated depletion of the residual 13 candidates on *ANKRD1* expression, a well-established direct YAP target gene (Fig. 1h)[12]. The depletion of several candidates led to the upregulation of *ANKRD1* expression after YAP 5SA induction compared to the cells transfected with a control siRNA. Using this approach, we identified TRPS1 as the best hit since depletion of this factor led to an approximately 20-fold super-induction of *ANKRD1* expression under doxycycline-induced conditions compared to the control siRNA. TRPS1 is a transcriptional repressor protein that contains several zinc finger motifs: C2H2, GATA and IKAROS-like zinc fingers (Fig. 1i). TRPS1 is able to repress target genes containing a GATA-binding site in their promoter and is commonly overexpressed in breast cancer compared to normal tissue[21,22]. Consequently, we considered TRPS1 to be a good candidate to restrain YAP target gene expression.

**TRPS1 represses YAP's transcriptional output**. To investigate the functional relationship between TRPS1 and YAP target gene expression, we used two breast cancer cell lines, MCF7 and T47D, expressing high levels of TRPS1 on protein level (Supplementary

Fig. 2a). First, we wanted to determine which genes are directly affected by the deregulation of YAP activity in this type of breast cancer cell lines. Since YAP mainly binds to enhancer regions, this cannot easily be deduced from ChIP-Sequencing data.

Therefore, we generated a MCF7 cell line, MCF7 i5SA, carrying a doxycycline-inducible Strep-YAP 5SA construct (pInducer-Strep YAP 5SA) and determined the shortest doxycycline induction time that is sufficient to induce YAP target gene

expression in order to limit it to direct YAP targets (Supplementary Fig. 2b, c). After induction of YAP 5SA expression for 14 h in MCF7 i5SA cells, we performed RNA-Sequencing and identified 497 significantly upregulated genes ($\log_2 FC > 1$, FDR $< 0.05$) (Fig. 2a, b). To evaluate the effect of TRPS1 depletion on this YAP-responsive gene set, we depleted TRPS1 in MCF7 cells using three independent shRNAs and by an siRNA approach in T47D cells, leading to efficient depletion of TRPS1 in both cell lines

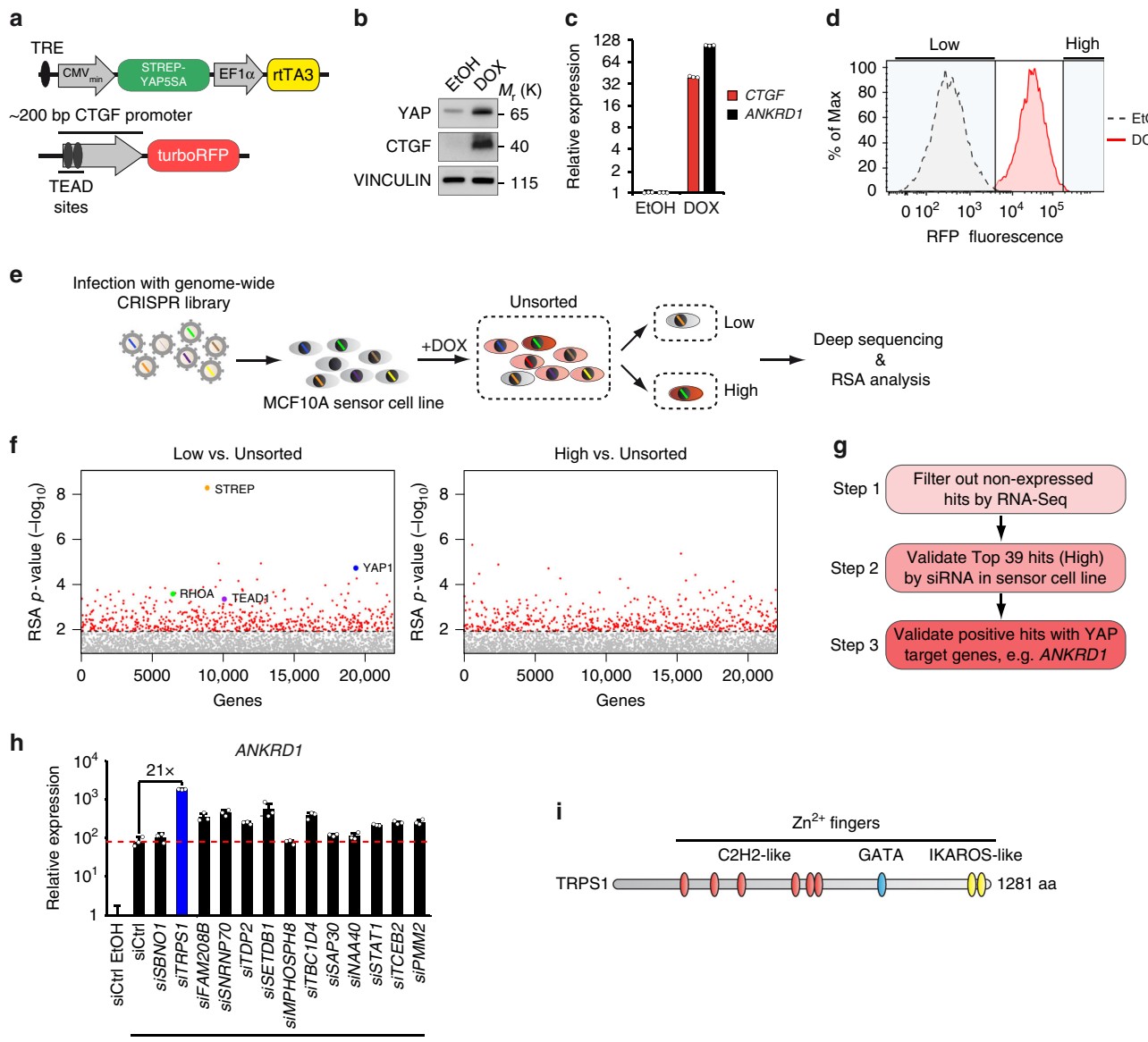

**Fig. 1** Identification of the YAP modulator TRPS1 using a genome-wide CRISPR screen. **a** Schematic of the YAP activity sensor system. The sensor cell line harbours a doxycycline inducible Strep-YAP5SA allele and a turboRFP (red fluorescent protein) reporter under the control of a *CTGF* promoter fragment containing TEAD-binding sites. **b** Western blot for YAP and CTGF in sensor cells treated with doxycycline (DOX) or ethanol (EtOH). Vinculin serves as loading control. **c** qRT-PCR analysis of the sensor cell line for the YAP target genes *CTGF* and *ANKRD1*. The cells were treated with doxycycline (DOX) or ethanol (EtOH). The chart summarizes three biological replicates. Error bars represent s.e.m. **d** Flow cytometry for RFP in the sensor cell line after treatment with doxycycline (DOX) or ethanol (EtOH), respectively. **e** Schematic of the CRISPR screening strategy. MCF10A sensor cells were infected with the genome-wide lentiviral GeCKO v2 CRISPR library. After doxycycline (DOX) treatment, cells were sorted into two subpopulations, "high" or "low", representing 1% of the cells with highest or lowest RFP signal, respectively. Both populations were then analyzed by deep sequencing to determine the frequency of each sgRNA. **f** Plots for the distribution of *P*-values in the low vs. unsorted (left) and high vs. unsorted (right) cells, respectively, after analysis by the redundant siRNA activity (RSA) algorithm. **g** Workflow for the validation of candidates from the CRISPR screen. **h** qRT-PCR analysis of *ANKRD1* expression in the doxycycline-treated sensor cell line transfected with siCtrl or siRNA targeting candidate YAP modulators. The cells were treated with doxycycline (+DOX) to induce YAP 5SA expression or ethanol (EtOH) as a control. Data presented are means from technical triplicates and error bars represent s.d. **i** Schematic of the TRPS1 protein

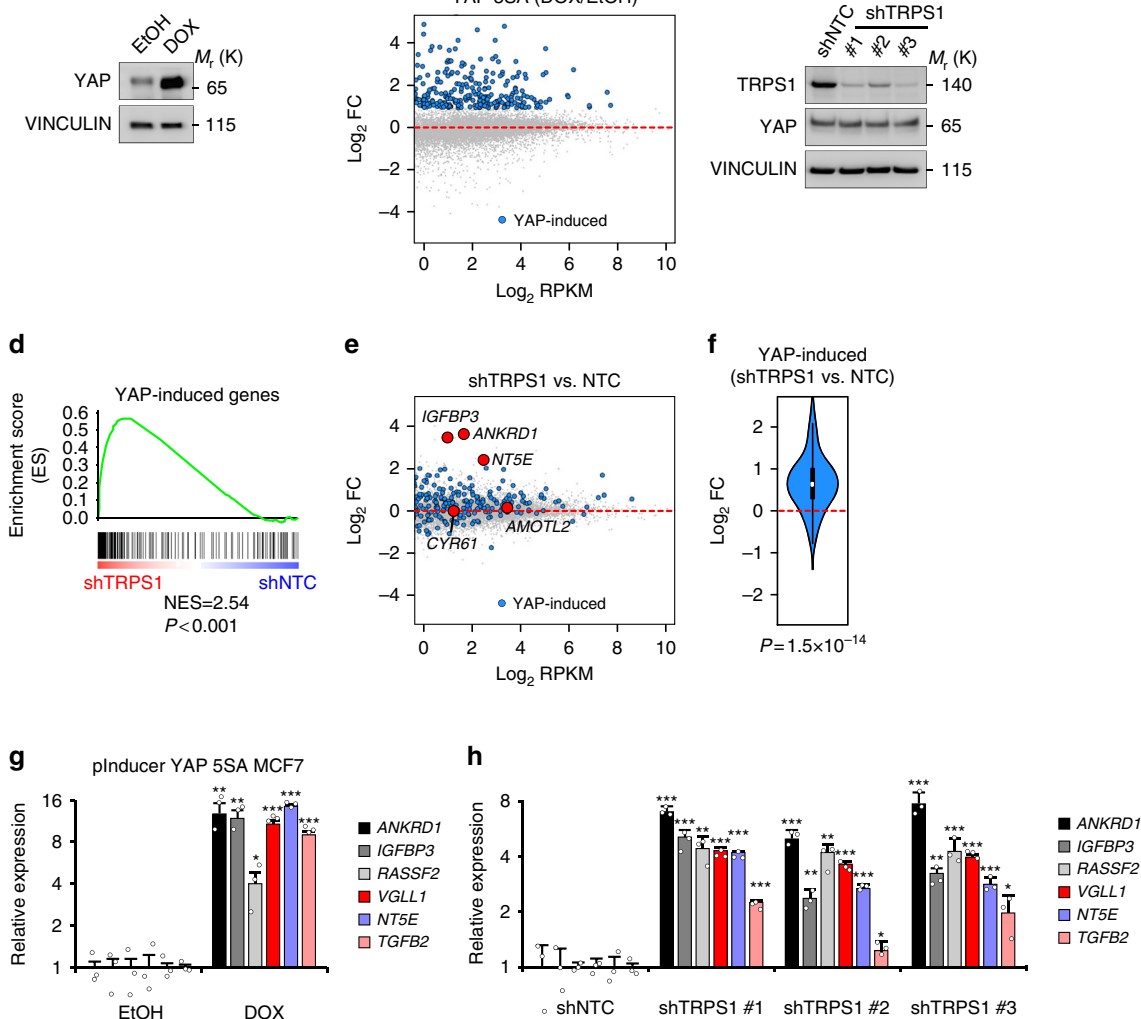

**Fig. 2** TRPS1 represses YAP target genes. **a** Western blot for YAP from MCF7 i5SA (pInducer-Strep YAP 5SA MCF7) cells after 14-h induction by doxycycline (DOX) or ethanol (EtOH), respectively. Vinculin serves as loading control. **b** MA plot for RNA-Sequencing from MCF7 i5SA cells after doxycycline (DOX) or ethanol (EtOH) treatment for 14 h. The $Log_2$ fold change ($Log_2FC$) is plotted against the abundance of transcripts given as number of reads per kilobase per million mapped reads (RPKM). YAP-induced genes are drawn as blue dots. **c** Western blot for TRPS1 and YAP after infection of MCF7 cells with three different TRPS1 shRNAs (shTRPS1 #1–3) or a non-targeting control (shNTC). Vinculin serves as loading control. **d** Gene set enrichment analysis (GSEA) for a set of 497 YAP-induced genes in RNA-Sequencing from TRPS1-depleted MCF7 cells. NES normalized enrichment score. **e** MA plot for RNA-Sequencing from MCF7 cells infected with three different TRPS1 shRNAs or a non-targeting control (shNTC). The $Log_2$ fold change ($Log_2FC$) is plotted against the abundance of transcripts given as number of reads per kilobase per million mapped reads (RPKM). YAP-induced genes are drawn as blue dots. Red dots label established YAP targets. **f** Violin Plot showing the differential regulation of YAP-induced genes after TRPS1 depletion by shTRPS1.The white point indicates the median. Two-sided Wilcox test. **g** qRT-PCR analysis of several YAP target genes in MCF7 i5SA cells after induction of YAP 5SA expression by doxycycline (DOX) or ethanol control (EtOH). Data presented are means from three biological replicates ($n = 3$) and error bars represent s.e.m; *$P < 0.05$; **$P < 0.01$; ***$P < 0.001$; Student's $t$-test. **h** qRT-PCR analysis of the YAP target genes from **g** in MCF7 cells infected with three different TRPS1 shRNAs or a non-targeting control (shNTC). Data presented are means from technical triplicates per shRNA and error bars represent s.d.; *$P < 0.05$; **$P < 0.01$; ***$P < 0.001$; Student's $t$-test

(Fig. 2c, Supplementary Fig. 2d). We subsequently performed RNA-Sequencing in both cell lines and determined the effect of TRPS1 depletion on the set of 497 YAP-induced genes. TRPS1 depletion led to a significant induction of this YAP gene set as shown by gene set enrichment analyses (Fig. 2d, Supplementary Fig. 2e) and to a strong upregulation of well-established YAP target genes such as *ANKRD1, NT5E* or *IGFBP3*[23]. Interestingly, the expression of other known YAP targets, e.g. *CYR61* or *AMOTL2*, was barely affected (Fig. 2e–h Supplementary Fig. 2f). Thus, TRPS1 represses the expression of a large set of YAP target genes in breast cancer cells, but some target genes are affected to a lesser extent or are even spared from this repressive effect.

**TRPS1 and YAP/TEAD share an overlapping set of genomic sites**. To gain more mechanistic insights into TRPS1's repressive effect on YAP/TEAD target gene expression in breast cancer cells, we performed ChIP-Sequencing for TRPS1 in MCF7 and T47D cells, since genome-wide binding profiles for this factor had not been reported previously (Fig. 3a, b; Supplementary Fig. 3a, b). To this end, we generated a highly specific TRPS1 antibody giving a clear nuclear signal in immunofluorescence stainings, which was strongly decreased after transfection of TRPS1-specific siRNAs in MCF7 cells (Supplementary Fig. 3c, d). Additionally, we performed ChIP-Sequencing for TEAD1 in both cell lines and ChIP-Sequencing for YAP in MCF7 cells (Fig. 3a, b; Supplementary

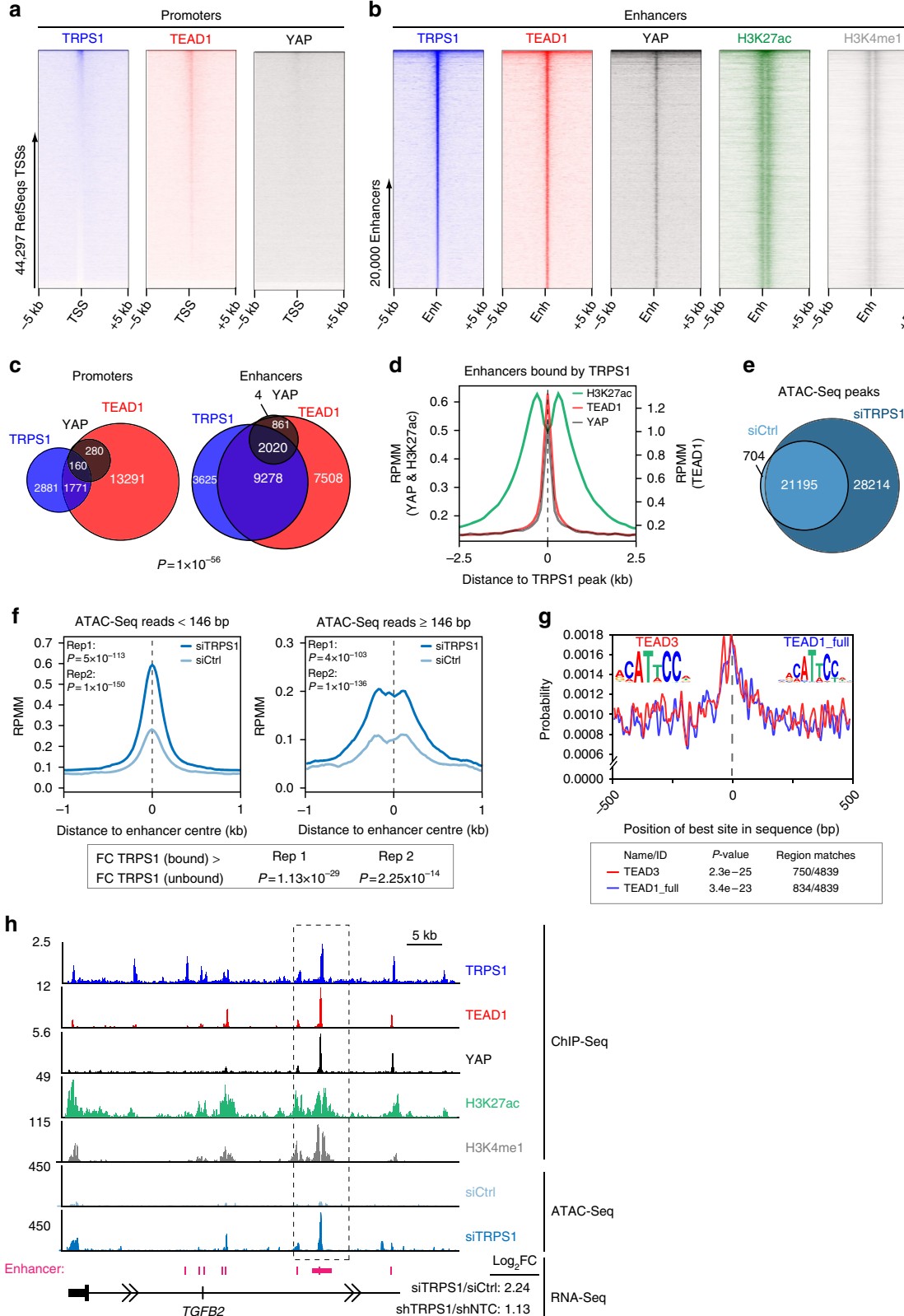

Fig. 3a). Due to the fact that YAP gets recruited to DNA via protein-protein interactions, and is therefore harder to capture by chromatin-immunoprecipitation, we performed ChIP-Sequencing for YAP in MCF7 i5SA cells after induction of YAP 5SA by doxycycline (Fig. 3a, b). As expected, induction of YAP 5SA led to a strong recruitment of exogenous YAP to the *CTGF*

promoter, thereby verifying the functionality of this system (Supplementary Fig. 3e). Since TEAD and YAP/TAZ preferentially bind to active enhancers[12,13], we first annotated active enhancer sites in MCF7 and T47D cells based on publically available ENCODE data sets where active enhancers are defined by the enhancer-specific chromatin mark acetylated

**Fig. 3** TRPS1 and YAP/TEAD bind to an overlapping set of genomic sites. **a** Heatmaps of ChIP-Seq data from MCF7 cells showing the occupancy of TRPS1, TEAD1 and YAP at all RefSeq transcriptional start sites (TSSs). The heatmap was sorted according to TRPS1 binding. **b** Heatmaps of ChIP-Seq data from MCF7 cells showing the occupancy of TRPS1, TEAD1 and YAP at all enhancer regions. The heatmap was sorted according to TRPS1 binding. The enhancer-specific chromatin marks were taken from a previously published data set[45]. The same contrasts for promoters (**a**) and enhancers (**b**) were used to demonstrate the differences in binding strength between the two. **c** Venn diagram showing the numbers of promoters and enhancers, respectively, bound by TRPS1, TEAD1 and YAP in MCF7 cells. **d** ChIP-Seq density profiles for H3K27ac, YAP and TEAD1 in a ±2.5-kb window surrounding the centre of TRPS1 peaks at enhancer sites. **e** Venn diagram for the numbers of ATAC-Seq peaks in T47D cells transfected with control siRNAs (siCtrl) or a siRNAs targeting TRPS1, respectively. **f** Distribution of ATAC-Seq reads with an insert size ≥146 bp (left), or with an insert size <146 bp (right) at enhancers bound by TRPS1 after TRPS1 depletion by siRNAs in T47D cells. *P*-values were calculated using a two-sided Wilcox-test. Rep1 replicate 1, Rep2 replicate 2. Indicated below are *P*-values for ATAC-Seq signals that describe if enhancers bound by TRPS1 are more strongly affected by TRPS1 depletion than enhancers not bound by TRPS1. Two-sided Wilcox-test. **g** Centrimo analysis for TEAD1 and TEAD3 binding motifs at TRPS1-bound enhancer sites in a 1-kb window. The reads are centred on the respective TRPS1 peak. **h** Sequencing tracks for ChIP-Seq data and ATAC-Seq data of the *TGFB2* locus. The last row shows the Log2 fold change (Log$_2$FC) of *TGFB2* expression determined by RNA-Seq in MCF7 and T47D cells after TRPS1 depletion and in MCF7 cells after overexpression of YAP 5SA

lysine 27 in histone H3 (H3K27ac) and/or by hypersensitivity to DNAse I.

The analysis of our ChIP-Sequencing data for TRPS1, TEAD1 and YAP revealed that TRPS1 binds more frequently to enhancers than to promoters and that, as previously reported, YAP/TEAD complexes follow the same trend (Fig. 3a, b)[12,13]. Depletion of TRPS1 by siRNAs led to strongly decreased recruitment of TRPS1 at eight randomly chosen TRPS1 binding sites identified in our ChIP-Sequencing experiments, further corroborating the specificity of our self-made TRPS1 antibody (Supplementary Fig. 3f).

The overlap of TRPS1 and YAP/TEAD sites was significantly better at enhancers ($P = 1 \times 10^{-56}$) compared to promoters (Fig. 3c; Supplementary Fig 3b). TRPS1 demonstrated a very good, though not perfect, overlap with YAP/TEAD complexes, with 2020 sites bound by TRPS1 out of 2881 (~70%) YAP/TEAD sites, which is consistent with TRPS1's specific effect on YAP target gene expression. Furthermore, the TEAD1 signal demonstrated a very narrow spatial distribution around TRPS1 peaks at enhancers compared to the bimodal signal for H3K27ac surrounding the signal for TEAD1 and TRPS1 (Fig. 3d). The expression of genes containing a TRPS1 peak within a 50-kb window of their promoter was significantly induced, indicating that TRPS1 acts mainly as a repressor (Supplementary Fig. 3g). Consequently, we hypothesized that TRPS1 might locally condense the chromatin structure to repress gene expression and enhancer function. To test this, we performed "Assay for Transposase-Accessible Chromatin with high throughput sequencing" (ATAC-Seq) experiments in T47D cells after TRPS1 depletion by siRNAs (Fig. 3e–g). Depletion of TRPS1 in T47D cells led to 28,214 additional significant ATAC-Seq peaks compared to the control-depleted cells, whereas only 704 additional sites occurred in the cells transfected with control siRNAs (Fig. 3e). In ATAC-Seq experiments performed with paired-end sequencing, reads can be stratified based on insert size. Reads generated from inserts smaller than the mononucleosome length (146 bp) are derived from nucleosome-free regions (NFRs). Depletion of TRPS1 led to a significant increase of the ATAC-Seq signal at TRPS1-bound enhancers for reads derived from inserts longer and shorter than 146 bp (Fig. 3f). Strikingly, these sites were centrally enriched for TEAD-binding motifs (Fig. 3g), indicating that chromatin becomes more accessible at TEAD-binding sites after TRPS1 depletion.

The increased chromatin accessibility was more pronounced at enhancers harbouring a TRPS1 ChIP-Seq peak in T47D cells compared to enhancers that did not show TRPS1 binding, implying that TRPS1 directly represses these sites (Fig. 3f). These results show that TRPS1 is locally compacting chromatin and is able to interfere with the establishment of NFRs. Thereby, TRPS1 is restricting the function of YAP/TEAD complexes at joint TRPS1-YAP/TEAD sites

as exemplified for an enhancer in the *TGFB2* gene body, a gene whose expression is induced by depletion of TRPS1 and YAP 5SA overexpression, respectively (Fig. 3h).

**TRPS1/TEAD bind to joint sites in a cooperative manner.** Our ChIP-Seq data sets revealed that TRPS1 and YAP/TEAD1 target a common set of enhancers and promoters where all three factors bind in a very close proximity, as shown exemplarily for the *VTCN1* enhancer (Chr1: 117,810,035 – 117,813,186) (Fig. 4a). To gain a better understanding of how TRPS1 selectively binds to a large set of YAP/TEAD1 sites but not to all YAP/TEAD1 sites, we had a closer look at the DNA motifs present in the ChIP-Seq peaks of TRPS1. GATA motifs were highly enriched in TRPS1 peaks, indicating that TRPS1 gets, at least partially, recruited to DNA via its GATA-like zinc finger (Fig. 4b). The analysis for GATA and TEAD-binding motifs within a ±250-bp window surrounding the top 500 TRPS1 peaks revealed that GATA motifs are significantly enriched in the centre of TRPS1 peaks, whereas TEAD1-binding motifs are more abundant in the vicinity of TRPS1 peaks (Fig. 4c). This finding suggests that specific sites for TEAD and TRPS1 have coevolved so that these genomic regions can be bound by both transcription factors at the same time. We hypothesized that the recruitment of TRPS1 and TEAD factors to joint sites might be stabilized by a protein–protein interaction between TRPS1 and TEAD factors in addition to the interactions between their DNA-binding domains and their cognate DNA sequences. We therefore analyzed the complex formation between TRPS1 and TEAD1 by proximity ligation assays (PLAs) in MCF7 cells (Fig. 4d). These analyses revealed a nuclear interaction between TRPS1 and TEAD1 and TEAD1 and YAP as a positive control. The PLA signal for the TRPS1–TEAD1 interaction was significantly reduced after TRPS1 depletion by siRNA, demonstrating the specificity of the assay (Fig. 4d, e). To corroborate the results from the PLA, we performed exogenous co-immunoprecipitation experiments between TRPS1 and the four Gal4-tagged TEAD transcription factors TEAD1–4 (Fig. 4f). TRPS1 was able to interact with all four TEAD transcription factors. We furthermore performed endogenous co-immunoprecipitation experiments for TRPS1 and TEAD factors (Fig. 4g). TEAD factors co-immunoprecipitated in TRPS1 immunoprecipitates from MCF7 cell lysates, which was not the case in lysates from TRPS1 knockout MCF7 cells, a cell line we generated by CRISPR/Cas9 technology (Fig. 4g, Supplementary Fig. 4a–c).

To test if TRPS1 is able to repress YAP/TEAD target gene repression in cis, we performed reporter assays using the *ANKRD1* promoter since this promoter is strongly bound by TRPS1 and YAP/TEAD1 (Supplementary Fig. 5a, b). In combination with TEAD1, YAP 5SA was able to potently transactivate the *ANKRD1* reporter construct (Supplementary Fig. 5c). TRPS1 was able to repress TEAD1/YAP 5SA activity in a dose-dependent manner

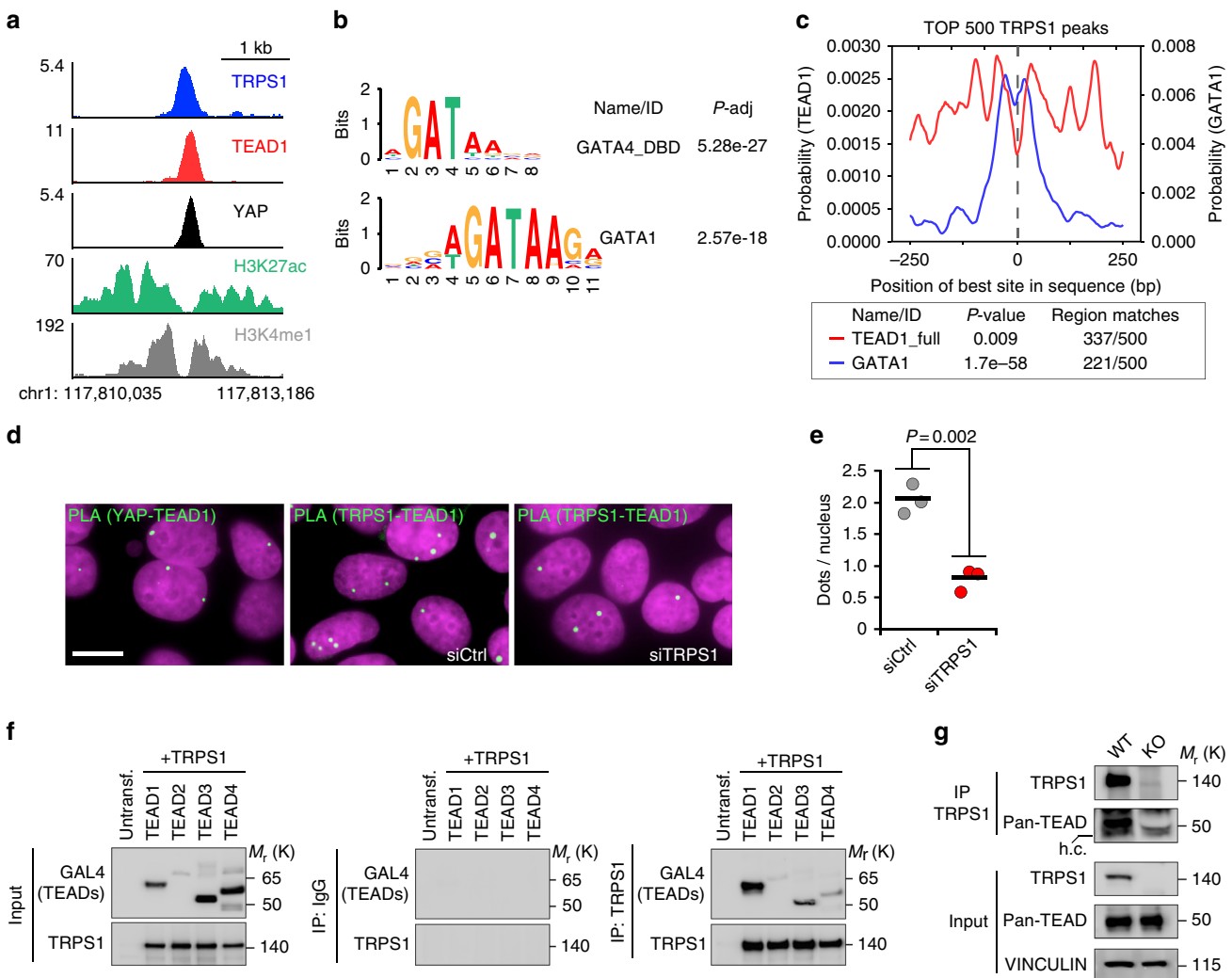

**Fig. 4** TRPS1 gets recruited to TEAD sites by cooperative binding. **a** ChIP-Seq tracks for TRPS1, TEAD1, YAP, H3K27ac and H3K4me1 at the *VTCN1* Enhancer region. **b** Motif enrichment analysis (DREME) for GATA4 and GATA1 binding sites in TRPS1 ChIP peaks. **c** Centrimo analysis for GATA1 and TEAD1 motifs to identify enriched binding sites surrounding the Top 500 TRPS1 ChIP-Seq peaks. **d** Representative pictures from proximity ligation assay (PLA) experiments in MCF7 cells. The pictures show PLA signals (in green) for YAP and TEAD (positive control, left) or TEAD1 and TRPS1 after treatment with siRNAs targeting TRPS1 or control siRNAs (siCtrl). **e** Quantification of PLA signals for the TEAD1-TRPS1 interaction in MCF7 cells treated with siTRPS1 or siCtrl, respectively. Values are presented as the number of dots per nucleus for each of the three biological replicates. A horizontal bar indicates the average. For quantification, at least 400 cells per replicate were counted. Student's *t*-test. **f** Co-immunoprecipitation experiment from 293T cells co-expressing TRPS1 and the indicated GAL4-tagged TEAD proteins. The lysates were subjected to immunoprecipitations using a TRPS1 antibody or IgG as a control and subsequently analyzed by Western blot. IgG controls and TRPS1 precipitates were analyzed on the same membrane. **g** Endogenous co-immunoprecipitation experiment from MCF7 WT and KO TRPS1. The lysates were subjected to immunoprecipitations using **a** TRPS1 antibody and subsequently analyzed by Western blot using a Pan-TEAD antibody; h.c. heavy chain

(Supplementary Fig. 5c). The TRPS1-mediated repression was dependent on the presence of GATA motifs in the *ANKRD1* promoter, since TRPS1 was not able to repress a truncated *ANKRD1* promoter lacking GATA sites (*ANKRD1* ΔGATA) (Supplementary Fig. 5d). Thus, the ability of TRPS1 to repress YAP/TEAD-dependent target sites is probably dictated by at least two factors: first, a GATA-binding site located in close proximity to a YAP/TEAD site; second, a protein–protein interaction between TRPS1 and TEAD transcription factors.

**TRPS1 recruits corepressor complexes to chromatin.** To decipher the precise biochemical mechanism by which TRPS1 represses YAP/TEAD activity, we set out to identify the interactome of TRPS1 through proximity-dependent biotin identification (BioID) experiments (Fig. 5a). This method is based on the

ability of the hyperactive BirA R118G (abbreviated BirA*) mutant to convert biotin to bioAMP, a highly reactive compound which biotinylates the lysines of all proteins located within a 10-nm radius[24], enabling their isolation and identification. We expressed the BirA*-Flag-TRPS1 protein in 293T cells and as a control, a BirA* protein containing a nuclear localization signal (NLS-BirA*-Flag) allowing the elimination of artefacts that might arise from a difference of compartmentalization between the control and TRPS1 protein. We first verified that BirA*-Flag-TRPS1 and NLS-BirA*-Flag were both able to induce nuclear biotinylation of target proteins in a time-dependent manner upon biotin addition (Supplementary Fig. 6a, b). After the expression of BirA*-Flag-TRPS1 or NLS-BirA*-Flag, cells were treated with biotin for 18 h, biotinylated proteins were pulled down with streptavidin beads and identified by mass spectrometry (Fig. 5a, b). This procedure was highly reproducible since the results from three independent

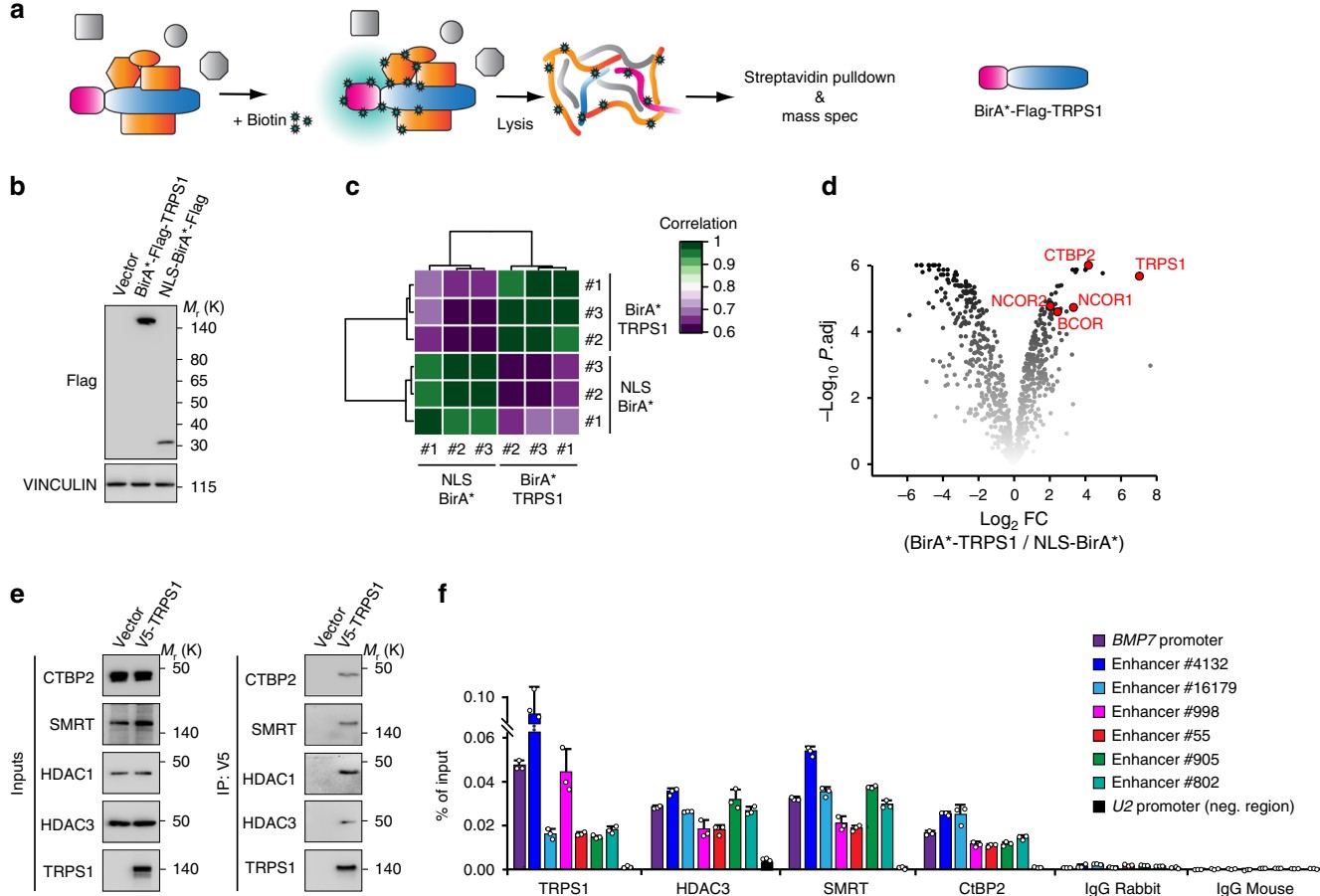

**Fig. 5** TRPS1 recruits corepressor complexes to chromatin. **a** Scheme depicting the BioID method. Mutant BirA (BirA*) fused to TRPS1 biotinylates all proteins in close proximity to the fusion protein after addition of biotin. **b** Western blot for BirA*-Flag-TRPS1 and the NLS-BirA*-Flag control from transfected 293T cells. Vinculin serves as loading control. **c** Heatmap clustering analysis (k-nearest neighbour) for the biological triplicates of the BioID experiment. **d** Volcano plot for nuclear proteins detected in all three BioID replicates. Log2FC: Log2 fold change; *P*. adj: adjusted *P*-value. **e** Western blot for CTBP2, SMRT, HDAC1 in TRPS1 co-immunoprecipitates from 293T cells expressing V5-TRPS1. **f** qChIP analysis for TRPS1, CTBP2, SMRT and HDAC3 binding at several enhancers and at the *BMP7* promoter. The *U2* promoter served as a control region. Control IPs were performed with rabbit or mouse IgG, respectively. Error bars represent s.d.

biological experiments showed a very good correlation between the replicates and cluster analysis (k-nearest neighbour) showed a clear separation between BirA*-Flag-TRPS1 and NLS-BirA*-Flag samples (Fig. 5c). As expected from the very strong auto-biotinylation observed in immunoblot analyses, TRPS1 was identified as the best hit (Fig. 5d, Supplementary Fig. 6a). Interestingly, several well-established corepressor complexes, e.g. CTBP2, NCOR1, NCOR2 (SMRT) and BCOR, were among the most significantly enriched proteins. We validated the identified interaction partners in subsequent co-immunoprecipitation experiments using V5-tagged TRPS1 (Fig. 5e). Corepressor proteins CTBP2, SMRT (NCOR2) and their associated histone deacetylases HDAC1 and HDAC3, were specifically detected in the V5-TRPS1 precipitates. Consistent with these results, we were able to detect CTBP2, SMRT and HDAC3 at enhancer sites and the *BMP7* promoter, which are bound by TRPS1 (Fig. 5f). Hence, we concluded that TRPS1 restrains YAP/TEAD-dependent transactivation by recruiting different corepressor complexes to joint genomic sites.

**TRPS1 regulates chromatin and long-range interactions**. Based on the results of the BioID experiments, we hypothesized that TRPS1 recruits several corepressor complexes, which

subsequently modify the chromatin to repress the activity of target enhancers. To verify this hypothesis, we performed ChIP-Sequencing experiments for CTBP2 and HDAC3 in MCF7 cells since these two factors have previously been implicated in the regulation of enhancers by deacetylating H3K27ac, a hallmark of active enhancers[25,26]. Similar to the binding preferences of TRPS1 and YAP/TEAD, both factors preferentially bound to enhancers compared to promoters in MCF7 wild-type cells (Fig. 6a). At enhancer sites, CTBP2 peaks, as well as HDAC3 peaks, demonstrated a highly significant overlap with the TRPS1 peaks that we identified in our previous experiments in MCF7 cells ($P = 4.28 \times 10^{-268}$ for CTBP2; $P = 6.08 \times 10^{-53}$ for HDAC3). Around 85% (8288/9754) CTBP2 peaks and 97% (593/614) HDAC3 peaks overlapped with a TRPS1 peak at enhancers (Fig. 6b). To determine how a complete loss of TRPS1 expression might affect the recruitment of cofactors and chromatin per se, we performed additional ChIP-Seq experiments in MCF7 TRPS1 knockout cells (Fig. 6c, Supplementary Fig. 4a–c).

When comparing the binding profiles of CTBP2 and HDAC3 in MCF7 cells and *TRPS1* KO cells, CTBP2 showed a strongly and HDAC3 a moderately reduced binding in the absence of TRPS1 (Fig. 6d, e). Our ChIP-Sequencing for H3K27ac in MCF7 and *TRPS1* KO cells demonstrated that the deletion of TRPS1 leads to a significant increase of H3K27ac at its

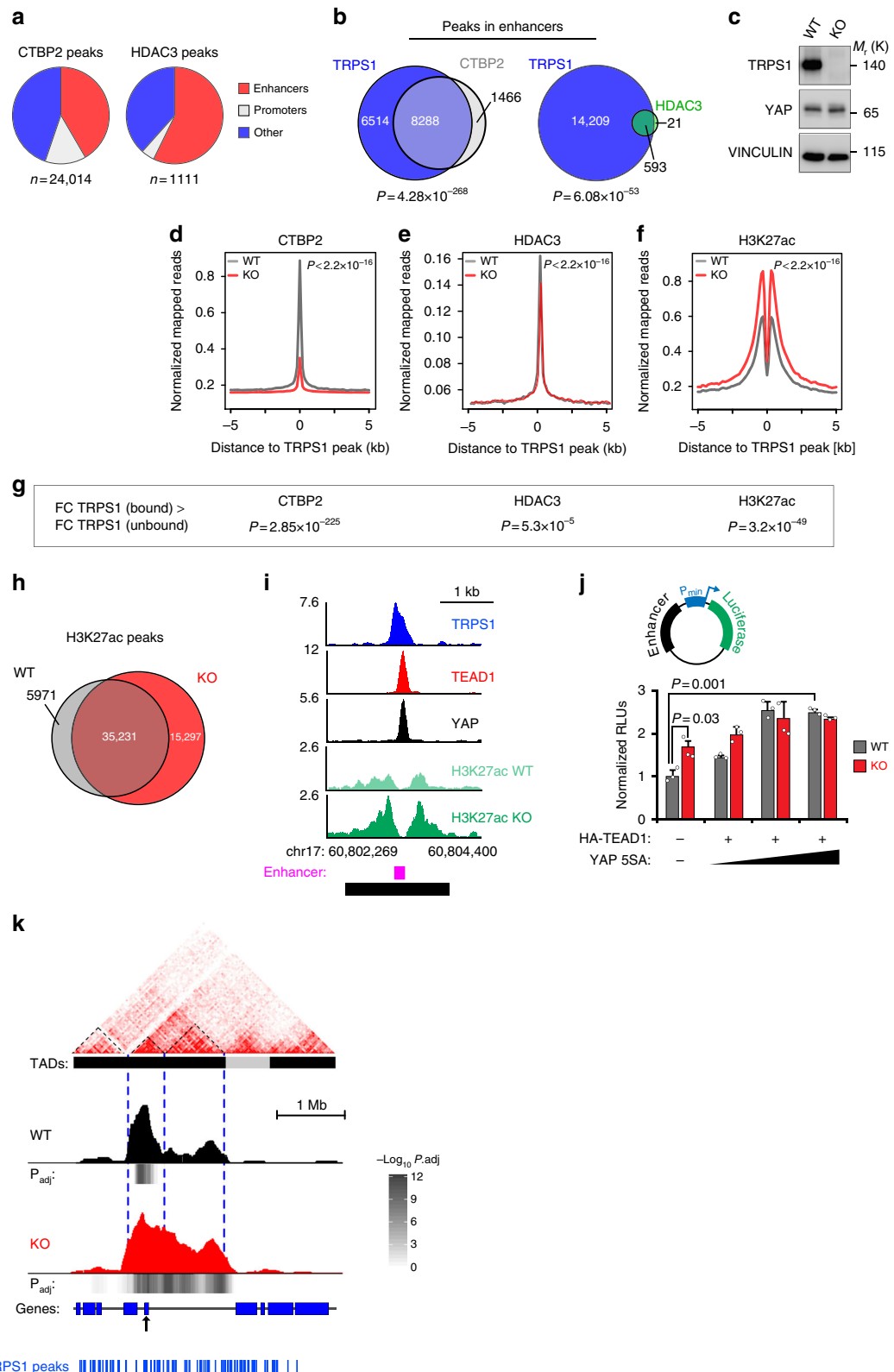

target sites (Fig. 6f). Consequently, the deletion of TRPS1 led to a significant upregulation of H3K27ac at enhancers bound by TRPS1, indicating that the recruitment of corepressors by TRPS1 is critical for this factor to restrain enhancer function.

The effect of TRPS1 deletion on CTBP2 and HDAC3 recruitment, as well as the increase in H3K27ac, was

significantly more pronounced in enhancers bound by TRPS1 compared to enhancers not bound by TRPS1, demonstrating that the observed changes are not due to universal changes at all enhancer sites (Fig. 6g). Moreover, TRPS1 deletion led to additional 15,297 H3K27ac peaks in *TRPS1* KO compared to WT MCF7 cells, whereas MCF7 WT cells only

**Fig. 6** Loss of TRPS1 alters chromatin structure and long-range interactions. **a** Pie charts depicting the proportion of enhancer and promoter regions among CTBP2 and HDAC3 peaks. **b** Venn diagrams showing the overlap of enhancer sites bound by TRPS1 and CTBP2 (left) or TRPS1 and HDAC3 (right). **c** Western blot for TRPS1 and YAP in MCF7 WT and *TRPS1* KO cells. **d**, **e** Density profiles for CTBP2 (**d**) and HDAC3 (**e**) binding at enhancers bound by TRPS1 in MCF7 and *TRPS1* KO cells. **f** Density profiles for H3K27ac at enhancers bound by TRPS1 in MCF7 WT and *TRPS1* KO cells. **g** P-values that describe if enhancers bound by TRPS1 are more strongly affected by TRPS1 deletion than enhancers not bound by TRPS1. Wilcox-test, two-sided. **h** Venn diagram showing the number of overlapping and unique H3K27ac ChIP peaks in MCF7 WT and *TRPS1* KO cells, respectively. **i** ChIP-Seq tracks for TRPS1, TEAD1, YAP and H3K27ac from MCF7 WT and *TRPS1* KO cells at an enhancer at the given genomic location. The annotated enhancer is drawn as a pink rectangle. The 2 kb fragment that was used for the luciferase assay in **j** is drawn as a black rectangle. **j** Luciferase activity of the reporter driven by the enhancer from **i** in MCF7 WT and *TRPS1* KO cells co-transfected with the indicated expression vectors. Increasing amounts of a FLAG-YAP 5SA construct and constant amounts of HA-TEAD construct were used. Data presented are derived from three biological replicates and error bars represent s.e.m. Student's *t*-test. **k** 4C-Seq interaction pattern of the *IGFBP3* TSS in MCF7 (WT) compared to TRPS1 KO (KO) cells. Significance was determined by the w4CSeq analysis package using a one-tailed binomial test and corrected for multiple testing yielding the adjusted P-value (P. adj). The HiC interaction map was used from a previously published ENCODE (ENCSR549MGQ) data set. Black and grey boxes mark TADs and a dashed line marks sub-TADs of the left TAD. The locations of TRPS1 ChIP-Seq peaks and annotated enhancers are given in blue and pink, respectively. TAD topologically associated domain

gained 5971 additional peaks compared to *TRPS1* KO cells (Fig. 6h).

Based on these results, we next set out to functionally test the repressive effect of TRPS1 on target enhancer output in a luciferase reporter assay (Fig. 6i, j). To do so, we selected one enhancer that contains peaks for TRPS1, TEAD1 and YAP, and demonstrates a strong increase in H3K27 acetylation after TRPS1 deletion (Fig. 6i). A 2-kb enhancer fragment was cloned in front of a minimal promoter in a luciferase reporter vector and we analyzed luciferase activity after transfection of MCF7 WT and *TRPS1* KO cells with this construct (Fig. 6j). The background activity of the reporter was significantly increased in *TRPS1* KO cells compared to MCF7 WT cells. Transfecting increasing amounts YAP 5SA led to a significant induction of the reporter activity and eventually high YAP 5SA concentrations were able to level out differences between MCF7 and *TRPS1* KO cells (Fig. 6j). This demonstrates that TRPS1 restrains YAP/TEAD activity at joint enhancer sites and that saturating concentrations of YAP are able to level out these differences.

Finally, we tested if chromatin changes upon TRPS1 deletion also affect another key feature of enhancers, namely their ability to form long-range interactions with promoters. Enhancer–promoter interactions mostly occur within so-called topologically associated domains (TADs), regions within which frequent interactions can be observed. Within a TAD, one can further identify certain sub-domains, also referred to as sub-TADs, which consist mainly of frequent interactions between enhancers and promoters. To this end, we performed 4C-Sequencing (4C-Seq) in MCF7 and TRPS1 knockout cells, an assay that analyses the looping of a specific genomic region, also called viewpoint, to distant sites (Fig. 6k).

As a viewpoint, we chose the transcriptional start site (TSS) of *IGFBP3*, a gene whose expression was potently induced after YAP overexpression or TRPS1 depletion, respectively (Fig. 2g, h). These 4C-Seq experiments revealed that *TRPS1* deletion led to a strongly altered looping of *IGFBP3*'s TSS in *TRPS1* KO cells compared to TRPS1-proficient cells, enabling interactions with additional sub-TADs. We were not able to correlate this effect with altered activity of a single enhancer, probably due to the fact that dozens of enhancers lie in the genomic region of the *IGFBP3* locus (Fig. 6k).

In summary, these experiments demonstrate that TRPS1 is locally modifying chromatin structure at enhancers, e.g. by deacetylating H3K27ac, ultimately leading to altered enhancer–promoter interactions.

**TRPS1 is overexpressed in breast cancer.** There have been a few reports stating that *TRPS1* is commonly overexpressed in breast cancer[22,27]. However, a recent study described that loss of *Trps1* expression could cooperate with decreased *Pten* expression in a transposon-based mutagenesis breast cancer screen in the mouse[28].

Thus, we analyzed the expression pattern of TRPS1 in sections of breast cancer samples and adjacent normal tissue (Fig. 7a; Supplementary Fig. 7a). Here, TRPS1 expression was confined to the luminal compartment in normal tissue. TRPS1 demonstrated a very strong nuclear staining in 27 out of 27 breast cancer patients, indicating that TRPS1 expression does not commonly get lost in breast cancer. This is supported by our analyses of the publically available TCGA data sets for breast cancer which show that the *TRPS1* locus is not deleted in human breast cancer patients but is rather amplified in a high fraction of patients leading to increased *TRPS1* mRNA expression (Fig. 7b, c). Amplification of *TRPS1* was associated with a substantially decreased survival probability of the affected patients, suggesting that *TRPS1* behaves as an oncogene in human breast cancer patients (Fig. 7d). Low YAP activity is associated with a poor survival for breast cancer patients[8], possibly due to the fact that YAP activation leads to activation of an anti-tumourigenic immunosurveillance response[4]. Thus, we hypothesized that TRPS1's oncogenic role could at least in part result from its repressive function on the transcriptional output of YAP. First, we determined the activity of TRPS1 and YAP, respectively, in the TCGA data set: we used gene signatures, which we generated based on our RNA-Sequencing data after YAP overexpression or TRPS1 depletion, respectively. We reasoned that low activity of TRPS1 in the tumours should lead to upregulation of its repressed genes from the RNA-Seq experiments, whereas high activity of YAP in the tumours should lead to upregulation of the genes identified after YAP 5SA overexpression. This analysis revealed that high TRPS1 activity is strongly associated ($\rho = 0.85$, $P < 2.2 \times 10^{-16}$) with low YAP activity and vice versa (Fig. 7e). The anti-correlation between TRPS1 and YAP activity was not confined to a specific breast cancer subtype (Fig. 7e). Strikingly, TRPS1 target genes were most strongly repressed in the Luminal B and Basal subtype in the TCGA data set (Fig. 7f), two rather aggressive subtypes. To validate these results, we performed multivariate analyses on an additional publically available data set[29]. This analysis included the expression of TRPS1-repressed target genes (defined by our RNA-Seq data) as readout of TRPS1 activity and several other clinical variables, e.g. lymph node-positivity or grade (Supplementary Fig. 7b, c; for all comparisons see Supplementary Data 1). Here, TRPS1 activity was an independent predictor of survival when all patients were included in the multivariate analysis (Supplementary Fig. 7b, c). High TRPS1 activity was also associated with decreased survival in lymph node-positive patients (Supplementary Fig. 7d, e) suggesting that TRPS1 independently contributes to the aggressiveness of breast cancer cells.

**TRPS1 is needed for efficient tumour growth in vivo.** To probe the oncogenic potential of TRPS1 in vivo, we made use of an orthotopic tumour model for basal breast cancer in which 4T1 cells, a BALB/C-derived tumour cell line, are transplanted

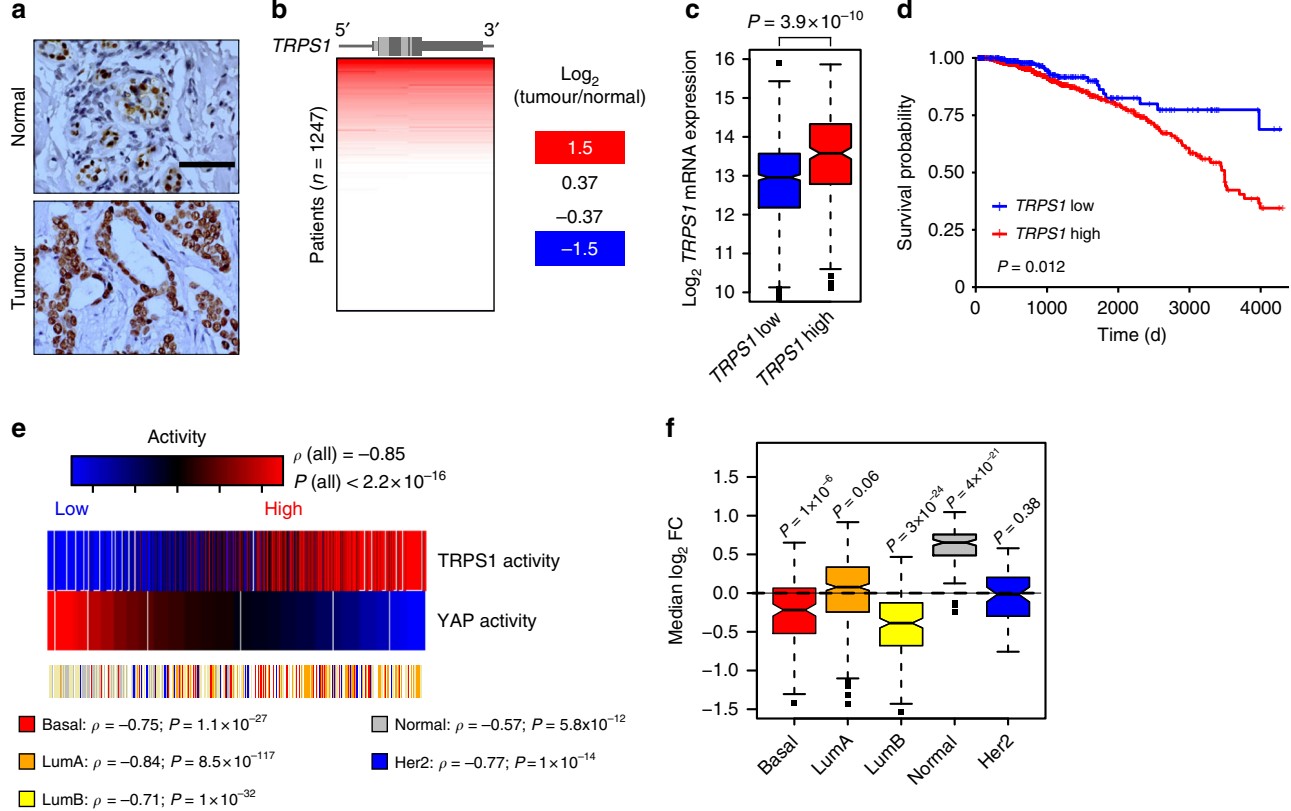

**Fig. 7** TRPS1 is overexpressed in breast cancer and is predictive for survival. **a** Immunohistochemical staining for TRPS1 on tissue sections from human breast cancer patients. Scale bar = 50 μm. **b** *TRPS1* gene amplification level in tumour vs. normal breast tissue from breast cancer patients of the TCGA data set. **c** Box plot showing *TRPS1* mRNA levels in patients stratified according to the level of amplification of the *TRPS1* locus. Median: black line; boxes: data points between the first and third quartiles; whiskers: up to 1.5 × interquartile range; points: outliers. Wilcox-test, two-sided. **d** Kaplan–Meier plot for the survival probability of breast cancer patients that were stratified based on their *TRPS1* amplification status. TRPS1 high: patients with *TRPS1* amplification status above the third quartile compared to TRPS1 low (remaining patients). Chi-square test. **e** Analysis of TRPS1 and YAP activity in breast cancer patients (see Methods). The correlation between TRPS1 and YAP activities is given for all patients (all) and for the given breast cancer subtypes. The correlation coefficients ($\rho$) and the corresponding P-values were determined by a Spearman rank correlation test. **f** Box plots showing the expression of genes repressed by TRPS1 in different breast cancer subtypes. Median: black line; boxes: data points between the first and third quartiles; whiskers: up to 1.5 × interquartile range; points: outliers. Wilcox-test, two-sided

into the mammary fat pad of syngeneic immunocompetent BALB/C mice and tumour growth is monitored over time. This tumour model was previously used to demonstrate the tumour suppressive function of YAP activation because high YAP activity elicits an anti-tumorigenic immune response[4]. 4T1 cells demonstrated elevated *Trps1* mRNA expression compared to mouse mammary epithelial cells (Supplementary Fig. 7f) suggesting that TRPS1 could have oncogenic functions in these cells.

To test this hypothesis, we depleted Trps1 in 4T1 cells using two potent shRNAs (Fig. 8a) and injected these cells into the mammary gland of BALB/C mice. Even though Trps1 depletion had no measurable effect on cell proliferation in vitro (Supplementary Fig. 7g, h), Trps1-depleted 4T1 cells demonstrated a strongly impaired tumour growth in vivo (Fig. 8b–d). Trps1 depletion was associated with a significantly increased frequency of intra-tumoural T cells as determined by immunohistochemical stainings for CD3 (Fig. 8e, f). Importantly, gene expression deconvolution of TCGA RNA-Sequencing also revealed a relationship between high Trps1 activity and low frequency of intra-tumoural immune cells (Fig. 8g) since tumours demonstrating high Trps1 activity showed a strongly reduced frequency of CD4, CD8 and natural killer (NK) cells, respectively (Fig. 8g).

Thus, TRPS1 is commonly overexpressed in human breast tumours, is anti-correlated with YAP activity and 4T1 cells

critically depend on high Trps1 expression to efficiently establish tumour formation in vivo, possibly due to TRPS1's effect on immunosurveillance.

## Discussion

Using a genome-wide CRISPR screening approach in combination with unbiased genomic and proteomic experiments, our study demonstrates that TRPS1 acts as a bona fide repressor of YAP/TEAD-dependent gene regulation and enhancer–promoter interactions. Consistent with an oncogenic role of TRPS1, its amplification is predictive for the survival of breast cancer patients, anti-correlated with YAP activity and with the frequency of tumour-infiltrating immune cells.

TRPS1 belongs to the family of GATA transcription factors. These factors are able to act as pioneer factors, meaning that they possess the striking capability to establish enhancer functions during development at genomic sites that cannot be accessed by other transcription factors. GATA1–6 behave as activators of transcription and enhancer function; in contrast, TRPS1 is the only member of the GATA family that can act as a transcriptional repressor and has the potential to antagonize enhancer function at GATA binding sites[21]. Here, we show that TRPS1 restrains H3K27ac at these GATA-binding sites, probably due to its ability to recruit corepressor proteins (Fig. 9). Whether the changes in

**Fig. 8** TRPS1 is needed for efficient tumour growth in vivo. **a** Western blot for TRPS1 in 4T1 cells infected with two different shRNAs targeting TRPS1 or a non-targeting shRen (Renilla) control, respectively. Serial dilutions (lanes 1–3) of the shRen control are loaded to demonstrate the knockdown efficiency. **b** Tumour growth of 4T1-derived tumours orthotopically transplanted into the mammary gland of BALB/C mice. Tumour size was monitored after the indicated times of injection. Data are presented as means ± s.e.m. Two-way ANOVA test. **c** The tumour weight of the 4T1-derived tumours was determined 23 days after transplantation. One-way ANOVA test. **d** Photos of 4T1-derived tumours 23 days after transplantation. Scale bar = 1 cm. **e** Representative images of IHC stainings for CD3-positive cells (T cells) in sections of the 4T1-derived tumours. Scale bar = 50 μm. **f** Quantification of (k). Error bars represent s.e.m. One-way ANOVA test. **g** Expression of signature gene sets for natural killers (NK), cytotoxic T cells (CD8) and helper T cells (CD4) were sorted based on TRPS1 activity in breast cancer patients. Spearman rank correlation test

enhancer–promoter interactions at the *IGFBP3* TSS in *TRPS1* KO cells are a direct consequence of the increased H3K27ac or whether it represents an additional TRPS1 function needs to be addressed in the future.

Interestingly, the fruit fly lacks a TRPS1 homologue. Thus, it seems that TRPS1 was created in the course of evolution to specifically shape the transcription of YAP/TEAD factors, probably in a tissue-specific manner. However, according to our ChIP-Seq studies, a significant number of TRPS1 binding sites are not co-occupied by TEAD1, likely reflecting additional YAP/TEAD-independent functions of TRPS1.

YAP activity is commonly downregulated during tumorigenesis in several tumour types, such as colon and breast cancer[3,8]. We demonstrated previously that deregulated MYC is also able to interfere with YAP/TEAD-dependent activity during breast cancer tumorigenesis[8,15,30]. This suggests that breast cancer tumours

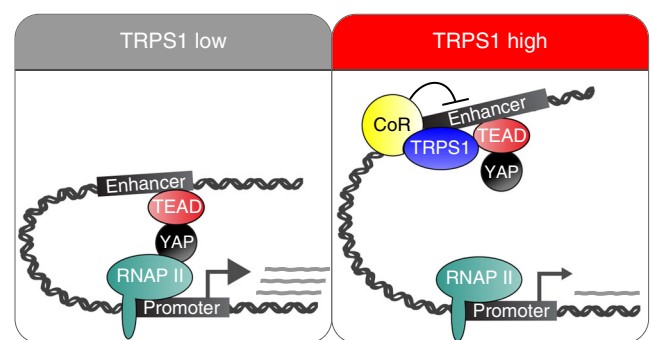

**Fig. 9** Model of how TRPS1 modulates YAP/TEAD-dependent transcription. See Discussion for details. CoR corepressors

have developed several strategies to repress YAP/TEAD activity. Strikingly, the loci for *TRPS1* (8q23.3) and *MYC* (8q24.21) are both located in close vicinity on chromosome 8 and are commonly co-amplified in human breast cancer[27] suggesting that these two factors cooperate during breast cancer development to efficiently shut down YAP/TEAD function. A similar phenomenon was previously shown for *PVT1* (a gene in the vicinity of the MYC locus), since *MYC/PVT1* co-amplification is required for efficient tumour formation in *MMTV-Neu*-driven tumours[31]. Although it is conceivable that tumour cells downregulate YAP activity to escape the immunosurveillance mechanisms, it is hard to reconcile this with YAP/TAZ ability to confer cancer stem cell traits to breast epithelial cells and its pro-oncogenic functions[32,33]. We propose that breast cancer cells need to maintain a certain level of YAP/TAZ activity that is high enough to maintain the pro-oncogenic functions of YAP/TAZ but sufficiently low to escape immunosurveillance. Thus, TRPS1 could exert its oncogenic role by keeping YAP/TAZ activity in check. In the future, it will be important to discriminate between YAP/TAZ-dependent and YAP/TAZ-independent functions for the oncogenic role of TRPS1 in vivo.

Up to now, the role of *Trps1* in the adult mouse could not be investigated since no conditional *Trps1* mouse model was available and *Trps1*-deficient mice die shortly after birth[34,35]. It will not only be important to uncover the role of TRPS1 during breast cancer, but also to understand its role during normal breast development and tissue maintenance using suitable mouse models. These studies will be instrumental in understanding if TRPS1 and/or its associated corepressor complexes might be useful targets for breast cancer therapy.

## Methods

**Tissue culture and transfection.** MCF7 and 293T/LentiX cells were cultivated in DMEM (+GlutaMAX, Thermo Scientific) supplemented with 10% FBS (Sigma) and 1% penicillin–streptomycin (Sigma). T47D culture medium was additionally supplemented with 1 μl/ml human insulin solution (Sigma). Cell lines were tested for mycoplasma contamination by PCR. MCF7, T47D and MCF10A cells were a kind gift from Almut Schulze (University of Würzburg, Germany) and they were authenticated using STR profiling.

Transfections were carried out using Lipofectamine 3000 reagent (Thermo Scientific) or polyethylenimine (PEI, Sigma) with Opti-MEM reduced serum medium (Thermo Scientific). For siRNA transfections, cells were transfected using the Lipofectamine RNAiMAX reagent (Thermo Scientific). siRNAs were purchased from Dharmacon and are listed in Supplementary Table 4.

**Lentiviral transduction.** LentiX (293T) cells were purchased from ATCC and used for lentivirus production. Cells were co-transfected with 10 μg psPAX2, 2.5 μg pMD2.G and 10 μg lentiviral vector using PEI (Sigma). Viral supernatants were harvested 24, 32, 48, and 56 h after transfection and pooled. For infection of target cells, the filtered viral supernatant was diluted with culture medium supplemented with 8 μg/ml protamine sulphate (Sigma). Infectious supernatant was removed after 24 h and selection of infected cells was started after 48 h.

**Mouse mammary epithelial cell isolation.** Primary mammary epithelial cells were isolated from 18-week-old virgin C57BL/6J mice by a 15-h digestion with gentle collagenase/hyaluronidase mix (StemCell Technologies) and subsequent digestion with trypsin and dispase/DNAase I. Stromal cells were sorted out by FACS using Biotin-labelled anti-CD31, anti-CD45 antibodies and streptavidin-APC. Mammary epithelial cells were purified using EpCAM-PE and CD49f-FITC antibodies.

**Western blotting.** Cells were lysed in RIPA buffer (50 mM Hepes pH 7.9, 140 mM NaCl, 1 mM EDTA, 1% Triton X-100, 0.1% Na-deoxycholate, 0.1% SDS) containing sodium pyrophosphate and protease inhibitor cocktail (Sigma). Lysates were cleared by centrifugation, separated on 8% Bis-Tris gels and transferred to a PVDF membrane (Millipore). Membranes were blocked with 5% skim milk powder in TBS, probed with primary antibodies diluted in 5% BSA in TBS and finally incubated with the appropriate horseradish peroxidase-coupled secondary antibodies. Visualization was performed using chemiluminescence HRP substrate (Immobilon Western, Millipore). Antibodies used are listed in Supplementary Table 1.

**Exogenous and endogenous co-immunoprecipitation.** For exogenous co-immunoprecipitation, 293T cells were transfected using PEI (Sigma). Cells were lysed in HEGN buffer (20 mM HEPES pH 7.9, 10% glycerol, 0.2 mM EDTA pH 8.0, 0.2% NP-40, 5 mM beta-glycerophosphate, 5 mM sodium pyrophosphate, 5 mM NaF, 120 mM KCl) containing protease inhibitor cocktail (Sigma). For endogenous co-immunoprecipitation, cell lysates were additionally lysed by sonication for 20 cycles (30 s on, 30 s off) in a Bioruptor (Diagenode). Cleared protein lysates were incubated with antibody-coupled Dynabeads (Thermo Scientific) overnight at 4 °C. Immunoprecipitates were washed with HEGN buffer, boiled with sample buffer and subjected to immunoblotting. For V5-TRPS1 immunoprecipitates, V5-TRPS1 was eluted by two consecutive elution steps with 1 mg/ml V5 peptide (Sigma) for 15 min at 37 °C.

**Immunofluorescence staining.** 293T cells were seeded in eight-well plastic chambers (Ibidi). Cells were fixed with 3.7% formaldehyde in PBS for 10 min at room temperature (RT). Fixed cells were washed and permeabilized for 10 min in PBS with 0.2% Triton-X-100. Cells were incubated in blocking buffer (5% BSA in PBS) for 20 min and cells were either incubated with streptavidin PE-Cy7 (eBioscience) or a primary antibody (diluted in blocking buffer) at 4 °C overnight. Cells were subsequently washed with PBS and, if required, incubated with a fluorescently labelled secondary antibody (Thermo) for 1 h at RT. After washing with PBS, nuclei were counter-stained with Hoechst.

**PLA.** MCF7 cells were transfected with siCtrl and siTRPS1 using RNAiMAX (Thermo Scientific) and seeded in 12-well chambers (Ibidi) 48 h post transfection; 72 h after transfection, cells were washed with PBS and fixed with 4% Formaldehyde in PBS for 5 min at RT. Fixed cells were permeabilized for 20 min at −20 °C using 100% MeOH. Cells were subsequently washed with PBS and PLA was performed using the Duolink PLA reagents (Sigma) following the manufacturer's protocol.

**Pooled lentiviral CRISPR screen.** The GeCKO v2 CRISPR library was obtained from Addgene and amplified using the supplied protocol. Illumina sequencing verified an even distribution of sgRNAs within one order of magnitude. The MCF10A sensor line was infected with the GeCKO v2 library using an MOI of 0.5 and a redundancy of 200. Cells were treated with doxycycline (0.5 μg/ml) 4 days after infection and ~70 Mio cells were sorted based on their RFP signal with an Aria III sorter after 2 days of doxycycline induction. Two populations were sorted: a "low" population representing the 1% least epifluorescent cells and a "high" population representing the 1% most epifluorescent cells. One further sample containing ~70 Mio cells was used as an unsorted sample. Genomic DNA was isolated using DNAzol (Thermo) according to the manufacturer's instructions. A first PCR reaction (18 cycles for the unsorted sample; 20 cycles for the sorted samples) was performed with external primers using 5 μg of genomic DNA. The PCR reactions for the respective samples were pooled and 5 μl of this pooled sample was used for a second PCR (24 cycles for the unsorted sample; 27 cycles for the sorted samples) with barcoding primers. Illumina sequencing was performed on an Illumina GAIIx sequencer using custom-designed sequencing primers. The primers are given in Supplementary Table 2.

**GeCKO screen analysis.** FASTQ files were trimmed to a length of 20 bp and aligned to a bowtie reference file containing all sgRNAs of the library. The reads per sgRNA were counted and a normalized (normalized reads per sgRNA: (reads per sgRNA + 1)/total reads for all sgRNAs in sample). The enrichment of a sgRNA comparing the sorted with the unsorted population was determined as follows: normalized reads (sorted)/normalized reads (unsorted). These enrichments were subsequently analyzed by the RSA algorithm (http://winzeler.ucsd.edu/supplemental/Konig NatureMethod-2007/RSA.html).

**RNA-Sequencing.** Total RNA was extracted using RNeasy® Mini Kit (Qiagen) with on-column DNaseI (Qiagen) digestion. RNA integrity was verified with the Agilent 2100 Bioanalyzer automated electrophoresis system (Agilent Technologies). mRNA was isolated using the NEBNext® Poly(A) mRNA Magnetic Isolation Module (NEB) and library preparation was conducted with the NEBNext® Ultra RNA Library Prep Kit for Illumina (NEB) with Dual Index Primers (NEBNext® Multiplex Oligos for Illumina, NEB). Size selection of the libraries for 270 bp was performed with Agencourt AMPure XP Beads (Beckman Coulter). Cycles for amplification of the cDNA were determined and conducted using qRT-PCR. Libraries were quantified with the Agilent 2100 Bioanalyzer automated electrophoresis system (Agilent Technologies) and subjected to Illumina Sequencing (HiSeq 2500).

**Chromatin-Immunoprecipitation (ChIP) and ChIP-Sequencing.** Cells were cross-linked with 1% formaldehyde for 10 min at RT. Nuclei were extracted using hypotonic buffer (20 mM Tris, pH 7.4, 2 mM MgCl₂, 5% glycerol) and lysed with ChIP lysis buffer (20 mM Tris pH 7.4, 150 mM NaCl, 1% NP40, 0.5% DOC, 0.1% SDS, 1 mM EDTA). Chromatin was sonified to obtain fragments of 200 bp. Equilibration of beads (Dynabeads, Thermo Scientific) with antibodies (2 μg for

ChIP, 10 μg for ChIP-Seq) was performed overnight, following immunoprecipitation of chromatin for 6 h. After extensive washing, bound chromatin was eluted using ChIP elution buffer (50 M Tris, pH 8.0, 1 mM EDTA, 1% SDS, 50 mM NaHCO₃) and de-crosslinked and RNase A/Proteinase K (Roth) digested overnight. DNA was purified by phenol/chloroform extraction. ChIP samples were analyzed by qChIP or subjected to library preparation according to the manufacturer's protocol (NEBNext Ultra II DNA Library Prep Kit for Illumina, NEB) using Dual Index Primers (NEBNext Multiplex Oligos for Illumina, NEB). The libraries were sequenced on an Illumina HiSeq 2500. Antibodies used for ChIP/ChIP-Seq are listed in Supplementary Table 1.

**Quality filtering and alignment for RNA- and ChIP-Sequencing.** Adapter removal, size selection (reads >25 nt) and quality filtering (Phred score >43) of FASTQ files was performed with cutadapt (http://cutadapt.readthedocs.io/en/stable/guide.html#). Reads were then aligned to the human genome (hg19) using bowtie2 (2.2.9) using default settings. For ChIP-Sequencing, the --local option in bowtie2 was used.

**ChIP-Seq bioinformatics analyses.** For peak calling, we used MACS2 (2.1.1) with default settings. For generation of heatmaps, we used NGSplot (https://github.com/shenlab-sinai/ngsplot). As coordinates for the promoters, the TSSs of all RefSeqs were used. As coordinates for the enhancers, a publically available data set in MCF7 cells (ENCODE Accession: ENCSR148VIV) was used. The BEDTools suite (v2.19.0) was used to infer overlaps of ChIP-Seq summits with enhancer and promoter regions, respectively.

**4C-Sequencing (4C-Seq).** 4C-Seq was performed as previously described with some modifications[36]. Briefly, 3 × 10⁶ MCF7 and TRPS1 KO cells were cross-linked with 1% (v/v) formaldehyde (Sigma) in culture medium for 10 min at RT. Cells were lysed in lysis buffer (10 mM Tris-HCl (pH 8.0), 10 mM NaCl) containing 0.5% (v/v) NP-40. Nuclei were digested with 800 U EcoRI (NEB) in Cutsmart buffer (NEB) supplemented with 0.3% SDS and 2% Triton X-100, overnight at 37 °C while shaking. The first ligation was carried out with 100 U T4 DNA ligase (Thermo Scientific) at 16 °C overnight. After decross-linking and phenol/chloroform extraction, 25 μg of the DNA was digested with 100 U DpnII in DpnII buffer (NEB) overnight at 37 °C while shaking. After a second ligation carried out with 200 U T4 ligase at 16 °C overnight, DNA was extracted with phenol/chloroform and purified with the ChIP DNA clean and concentrator™ kit (Zymo Research). About 150 ng DNA was used as a template to amplify by PCR-specific libraries for each selected enhancer and control regions using the primers listed in Supplementary Table 2 and the Phusion hot start II polymerase (Thermo Scientific) as recommended by the manufacturer. Successful reactions were pooled and purified with Agencourt AMPure XP Beads (Beckman Coulter). Equal amounts of specific libraries were combined and sequenced on an Illumina Hiseq 2500 with 75 cycles.

**Analysis of 4C-Seq data.** For 4C-Seq analyses, the publically available w4CSeq analysis pipeline was used (https://github.com/WGLab/w4CSeq) using default settings. Distal regions with an adjusted P-value <0.01 were considered to significantly interact with the *IGFBP3* TSS. HiC data heatmaps were plotted for comparison using previously published data sets available at ENCODE (ENCSR549MGQ).

**Generation of TRPS1 knockout cell lines by CRISPR.** A small guide RNA targeting exon 3 of TRPS1 (CTGCTCTTTGCGGAGACTTC) was cloned into pX461 in which the nickase Cas9 allele was substituted for the wildtype Cas9. MCF7 cells were transfected using Lipofectamine 3000 (Thermo Scientific). After 48 h, GFP⁺ cells were isolated by flow cytometry and single cells were seeded in 96-well plates. After expansion, potential *TRPS1* KO clones were identified by Western blot using antibodies recognizing the N-terminal (Abcam Ab209664) and C-terminal region of the protein, respectively. To verify the knockout and to identify the responsible mutations, genomic DNA was isolated and regions flanking the sgRNA target site were amplified by PCR. The amplicons were cloned into pJET (Thermo Scientific) and analyzed by Sanger sequencing. Sequencing revealed frameshifts leading to premature stop codons.

**ATAC-Seq.** The ATAC-Seq experiments were performed as described previously[37]. We performed two biological replicates per condition. Briefly, 50,000 cells were used per reaction. Nuclei were isolated after resuspension and centrifugation in Lysis buffer (10 mM Tris-HCl, pH 7.4, 10 mM NaCl, 3 mM MgCl₂, 1% (v/v) Igepal CA-630). The 50 μl transposase reaction with isolated nuclei (plus 25 μl TD, 2.5 μl TDE1 and 22.5 μl H₂O) was incubated at 37 °C for 30 min. DNA was purified using a MinElutePCR purification column (Qiagen). The transposed DNA fragments were preamplified by a first PCR reaction with five cycles containing barcoded Nextera PCR primers. The optimal number of cycles was determined by a SybrGreen qPCR reaction containing a 5-μl aliquot from the first PCR. The second PCR was then carried out with eight cycles and the libraries were first purified by MinElutePCR purification column (Qiagen) and then further

size-selected by AMPure XP beads to obtain libraries with a size distribution between 150 and 1000 base pairs.

**ATAC-Seq analysis.** For analysis, we used a publically available ATAC-Seq pipeline (https://github.com/kundajelab/atac_dnase_pipelines). For calling ATAC-Seq peaks, only peaks with an Irreproducibility Discovery Rate (IDR) <0.1 were considered significant.

**BioID affinity purification and preparation for liquid chromatography tandem mass spectrometry (LC-MS/MS).** 293T cells were transfected with BirA*-Flag-TRPS1 and NLS-BirA*-Flag control in triplicates using PEI (plasmids are listed in Supplementary Table 5). Twenty-four hours after transfection, cells were treated with 50 μM biotin (Sigma) for 18 h; 2 × 10⁷ cells were collected per sample, spun down, snap frozen in liquid nitrogen and stored at −80 °C till further use. The cell pellets were resuspended in 4.75 ml lysis buffer (50 mM Tris pH 7.5, 150 mM NaCl, 1 mM EDTA, 1 mM EGTA, 1% Triton, 1 mg/ml Aprotinin, 0.5 mg/ml Leupeptin, 250 U Turbonuclease (Accelagen), 1 mM PMSF, 0.1% SDS). The cells were lysed by incubation for 1 h at 4 °C rotating at 15 rpm and finally sonicated five times for 30 s at 4 °C. The lysate was spun 30 min at 4 °C at 17,000×g and the supernatants were merged. Streptavidin Sepharose High Performance beads (GE Healthcare Life Sciences) were equilibrated in PBS and lysine acetylated using 20 mM sulpho-NHS-acetate (Pierce, ThermoFisher) for 1 h at RT. Sulpho-NHS-acetate was quenched by adding 1 M Tris pH 7.5 and beads were extensively washed with PBS. We found that acetylation of lysines reduces the background signal of streptavidin-derived peptides following on beads digestion. The acetylated streptavidin beads were equilibrated in lysis buffer, added to the lysate, and incubated for 3 h at 4 °C rotating at 15 rpm. Samples were centrifuged for 2 min at 4 °C at 2000×g, beads were resuspended in 200 μl remaining supernatant and transferred to a Spin Column (Pierce, ThermoFisher). The beads were washed twice using lysis buffer and five times with wash buffer (50 mM ammonium bicarbonate (AmBic), pH 8.3). The beads were transferred to a fresh tube using three times 300 μl wash buffer, spun down at 2000×g for 5 min at 4 °C and resuspended in 200 μl remaining supernatant; 1 μg of trypsin (Mass Spectrometry Grade, Promega) was added and incubated at 37 °C for 16 h shaking at 500 rpm. Then 0.5 μg of trypsin was added and the on-bead digest continued for additional 2 h. The beads were transferred to a new Spin Column and the digested peptides were eluted with two times 150 μl 50 mM AmBic and elutions were merged (AmBic elution). Biotinylated peptides still bound to the beads were eluted twice with 150 μl of 80% ACN and 20% TFA and ACN/TFA elutions were merged. AmBic and acetonitrile/trifluoroacetic acid (ACN/TFA) elutions were dried down with the speed vacuum centrifuge, resuspended in 200 μl solvent A (0.05% formic acid in milliQ water) and sonicated five times 60 s. The desalting and clean-up of the samples was carried out with Waters Oasis® HLB μElution Plate 30 μm in the presence of a slow vacuum. In this process, the columns were conditioned with 3 × 100 μl solvent B (80% acetonitrile; 0.05% formic acid) and equilibrated with 3 × 100 μl solvent A. The samples were loaded, washed three times with 100 μl solvent A and then eluted into PCR tubes with 50 μl solvent B. The eluates were dried down with the speed vacuum centrifuge and dissolved in 50 μl 5% acetonitrile, 95% milliQ water, with 0.1% formic acid prior to analysis by LC-MS/MS.

**LC-MS/MS.** Peptides from AmBic and ACN/TFA elutions were analyzed independently and separated using the nanoAcquity UPLC system (Waters) fitted with a trapping (nanoAcquity Symmetry C₁₈, 5 μm, 180 μm × 20 mm) and an analytical column (nanoAcquity BEH C₁₈, 1.7 μm, 75 μm × 250 mm). The outlet of the analytical column was coupled directly to an Orbitrap Fusion Lumos (Thermo Fisher Scientific) using the Proxeon nanospray source. Solvent A was water, 0.1% formic acid and solvent B was acetonitrile, 0.1% formic acid. The samples (1 μl) were loaded with a constant flow of solvent A at 5 μl/min onto the trapping column. Trapping time was 6 min. Peptides were eluted via the analytical column with a constant flow of 0.3 μl/min. During the elution step, the percentage of solvent B increased in a linear fashion from 3 to 25% in 30 min, then increased to 32% in 5 more min and finally to 50% in a further 0.1 min. Total runtime was 60 min. The peptides were introduced into the mass spectrometer via a Pico-Tip Emitter 360 μm OD × 20 μm ID; 10 μm tip (New Objective) and a spray voltage of 2.2 kV was applied. The capillary temperature was set at 300 °C. The RF lens was set to 30%. Full scan MS spectra with mass range 375–1500 m/z were acquired in profile mode in the Orbitrap with resolution of 120000. The filling time was set at maximum of 50 ms with limitation of 2 × 10⁵ ions. The "Top Speed" method was employed to take the maximum number of precursor ions (with an intensity threshold of 5 × 10³) from the full scan MS for fragmentation (using HCD collision energy, 30%) and quadrupole isolation (1.4 Da window) and measurement in the ion trap, with a cycle time of 3 s. The monoisotopic precursor selection (MIPS) peptide algorithm was employed but with relaxed restrictions when too few precursors meeting the criteria were found. The fragmentation was performed after accumulation of 2 × 10³ ions or after filling time of 300 ms for each precursor ion (whichever occurred first). MS/MS data were acquired in centroid mode, with the Rapid scan rate and a fixed first mass of 120 m/z. Only multiply charged (2⁺–7⁺) precursor ions were selected for MS/MS. Dynamic exclusion was employed with maximum retention period of 60 s and relative mass window of 10 ppm. Isotopes

were excluded. Additionally only one data-dependent scan was performed per precursor (only the most intense charge state selected). Ions were injected for all available parallelizable time. In order to improve the mass accuracy, a lock mass correction using a background ion ($m/z$ 445.12003) was applied. Data acquisition was performed using Xcalibur 4.0/Tune 2.1 (Thermo Fisher Scientific).

**Data analysis**. For the quantitative label-free analysis, raw files from the Orbitrap Fusion Lumos were analyzed using MaxQuant (version 1.5.3.28)[38]. MS/MS spectra were searched against the Human Swiss-Prot entries of the Uniprot KB (database release 2016_01, 20198 entries) using the Andromeda search engine[39]. A list of common contaminants was appended to the database search. The search criteria were set as follows: full tryptic specificity was required (cleavage after lysine or arginine residues, unless followed by proline); two missed cleavages were allowed; oxidation ($M$), acetylation (protein N-term) and biotinylation ($K$) were applied as variable modifications, mass tolerance of 20 ppm (precursor) and 0.5 Da (fragments). The reversed sequences of the target database were used as decoy database. Peptide and protein hits were filtered at a false discovery rate of 1% using a target–decoy strategy[40]. Additionally, only proteins identified by at least two unique peptides were retained. The LFQ intensity values per protein (from the proteinGroups.txt output of MaxQuant) were used for further analysis. All comparative analyses were performed using R version 3.2.3. The R package *MSnbase*[41] was used to process proteomics data and perform data imputation using *imputeLCMD*. Missing values were imputed using a mixed strategy based on the definition of Missing At Random (MAR) and Missing Not At Random (MNAR) values. MNAR were defined for each pairwise comparison as values that were (i) missing in 3 out of 3, or 2 out of 3 biological replicates in one sample group, and (ii) present in all the 3 biological replicates in the second sample group. Because of their non-random distribution across samples, these values were considered as underlying biological difference between sample groups. MNAR values were computed using the method "MinDet" by replacing values with minimal values observed in the sample. MAR were consequently defined for each pairwise comparison as values that were missing in 1 out of 3 biological replicates per sample group. MAR values were imputed based on the method "knn" (k-nearest neighbours)[41]. All the other cases (e.g., protein groups that had less than two values in both sample groups) were filtered out because of the lack of sufficient information to perform robust statistical analysis. Data were quantile normalized to reduce technical variations. Differential protein abundance between control NLS-BirA* and BirA*-TRPS1 lines was evaluated using the *limma* package[42]. Differences in protein abundances were statistically determined using the Student's *t*-test with variances moderated by *limma*'s empirical Bayes method. False discovery rate was estimated using *fdrtool*[43].

**Luciferase assay**. For the *ANRKD1* reporter construct, 293T/LentiX cells were transfected with a pGL4-20 *luc2*/Puro (Promega) vectors containing a fragment of the *ANKRD1* promoter or a version with mutated GATA site using PEI reagent. All plasmids used are listed in Supplementary Table 5. Foty-eight hours post-transfection, cells were lysed with passive lysis buffer (Promega) and luciferase activity was measured in a Luminometer (Berthold Technologies). Equal amounts of a CMV-beta-Gal construct were always co-transfected and luciferase light units were normalized to beta-galactosidase activity.

For the enhancer reporter construct, 80,000 MCF7 WT and *TRPS1* KO cells were seeded into 24-well plates and co-transfected the following day with 3 μg of pGL4.23 *luc2*/MinP (Promega) containing a 2-kb fragment (chr17:60,803,393 - 60,803,563), 12 ng of pCMV-HA-TEAD1, 120 ng of pRL-CMV Renilla and increasing amounts of pLeGo-iG2-Puro-Flag-YAP 5SA (2, 16, 128 ng) using Lipofectamine 3000 (Thermo Scientific). All plasmids used are listed in Supplementary Table 5. Cells were harvested 48 h post-transfection and assayed for luciferase activity using the Dual Glo® (Promega) luciferase assay system according to the recommendations of the manufacturer. Luciferase activity was measured in a Luminometer (Berthold Technologies) and the relative Enhancer activity was calculated as the ratio of firefly luciferase activity to Renilla luciferase activity.

**Statistics and reproducibility**. All statistical tests were performed using R. Quantifications of CD3-positive cells in 4T1 tumours were performed in a blinded fashion (see below). When we applied *T*-test statistics or ANOVA analyses, the variance was assumed to be similar between each group. Here, a sample size of $n = 3$ was considered sufficient to detect changes ($\alpha = 0.05$, Power = 0.8, effect size >4 according to Cohen). Variation was always indicated using standard deviation unless stated otherwise. The statistical tests applied and the number of replicates are always given in the respective figure legend.

**qRT-PCR**. RNA was extracted with peqGOLD TriFast Reagent (Peqlab). First-strand cDNA synthesis was performed using M-MLV Reverse Transcriptase (Promega) and random hexamer primers (Sigma) according to standard procedures. PCR reaction was performed in technical triplicates using ABsolute qPCR SYBR Green Mix (Thermo Scientific). Gene expression was analyzed with a StepOnePlus™ Real-Time PCR System (Thermo Scientific). The expression values were normalized to *b2M* as housekeeping gene using the ddCt method. The used primers are listed in Supplementary Table 2.

**Analysis of breast cancer data sets**. All analyses were performed in the R (v 3.2.3) environment. Gene expression data, copy number variation (CNV) data and corresponding clinical data for TCGA were downloaded from the respective webpages (https://tcga-data.nci.nih.gov/tcga/ or http://xena.ucsc.edu/, respectively). The analysis of TRPS1 CNV was performed with the http://xena.ucsc.edu/ online tool. The enrichment of specific gene signatures per patient was performed using a GSEA-like algorithm as described previously using a Kolgomorov–Smirnov test for normality[8,44]. Multivariate analysis was performed with a publically available data set (http://co.bmc.lu.se/gobo/). As a gene set, we used genes that were significantly upregulated (Log₂FC >1, FDR <0.01) according to our own RNA-Sequencing experiments after TRPS1 depletion in MCF7 cells.

Deconvolution of gene expression profiles to deduce the relative contribution of invading immune cells to the overall gene expression profile was performed with the R package "CellMix" using the supplied gene signatures for the respective cell type.

**TRPS1 immunohistochemistry on human breast cancer sections**. Twenty-seven formalin-fixed and paraffin-embedded (FFPE) tissues with invasive breast carcinoma of no special type (NST) were obtained from the archives of the Institute of Pathology (University of Würzburg) and the Interdisciplinary bank of biomaterials and data Würzburg (IBDW). Immunohistochemistry for TRPS1 was performed using standard protocols (anti-TRPS1 antibody, ab209664, Abcam; dilution 1:8000; pretreatment with pressure cooking in citric acid pH 6.0).

**4T1 orthotopic transplantation into BALB/C mice**. 4T1 mouse mammary cancer cell lines were generated by lentiviral delivery to express two different shRNA constructs targeting TRPS1 (shTRPS #1, shTRPS #2) as well as one control shRNA construct targeting Renilla (shRen). Five mice per group were injected which we considered a suitable sample size for this experiment. All animal procedures were performed in accordance with UK Home Office regulations under project license PPL/70/8380. For orthotopic transplantations, $2 \times 10^5$ 4T1 tumour cells were resuspended in 50 μl growth factor-reduced Matrigel (Costar) and transplanted into the fourth mammary fat pad of 14-week-old female BALB/c mice on one flank. Tumour growth was assessed morphometrically using calipers, and tumour volumes were calculated according to the formula $V$ (mm³) = $L$ (major axis) × $W2$ (minor axis)/2. Upon termination of the experiment tumours were excised and weighed. Five mice were injected per construct. One mouse (shTRPS1 #2) did not show tumour induction and one mouse (shTRPS1 #1) developed a tumour in the peritoneum. Both mice were consequently excluded from the analysis.

**Immunohistochemistry of 4T1 tumours**. Endpoint tumours were fixed in 10% neutral buffered saline for 24 h. Antigen retrieval was performed using 10 mM Citrate buffer pH 6.0.

Quantification of CD3-positive cells in tumours was performed in a blinded fashion. To this end, at least five random pictures from tumours were taken and counted by a third person. At least 3000 cells per tumour were counted.

**4T1 in vitro proliferation assays**. For assessment of 4T1 cell proliferation using an IncuCyte Zoom machine (Essen BioScience), cells expressing shTRPS #1, shTRPS #2 or shRen, $1 \times 10^3$ cells were seeded into the wells of a 96-well plate in triplicate and confluency was measured over 150 h. For assessment of 4T1 cell proliferation by a cumulative growth curve, $8 \times 10^4$ 4T1 cells were seeded in a six-well plate. After 3 days, cells were counted and replated at the initial density. The assay was performed in independent triplicates.

**Data availability**. RNA-Seq, ChIP-Seq, ATAC-Seq and 4C-Seq data can be found online at the GEO repository (Accession ID: GSE107023 . BioID data are available via ProteomeXchange with identifier PXD009819.

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

## Acknowledgements

B.v.E. was supported by a grant from the Else Kröner-Fresenius Foundation. The Fritz Lipmann Institute (FLI) is a member of the Leibniz Association and is financially supported by the Federal Government of Germany and the State of Thuringia. D.E. has received a fellowship from the Leibniz Graduate School on Aging (LGSA). The DNA Sequencing, the Flow Cytometry and the Proteomics facilities of the FLI are gratefully acknowledged. We would like to thank Jason Carroll (Cambridge) and Jos Jonkers (Amsterdam) for sharing unpublished data. We would like to thank the members of the von Eyss lab, von Maltzahn lab and Kaether lab for helpful suggestions and Christin Ritter for excellent technical support. We would also like to thank Elmar Wolf and Julia von Maltzahn for critical reading of the manuscript. This work was supported by the Francis Crick Institute which receives its core funding from Cancer Research UK (FC001144), the UK Medical Research Council (FC001144), and the Wellcome Trust (FC001144).

## Author contributions

D.E., M.T. and B.v.E performed the experiments and wrote the manuscript. B.v.E performed all computational analyses. K.S. designed and performed the orthotopic 4T1 experiments under supervision of E.S., A.R. performed TRPS1 immunohistochemistry on patient samples. A.O. analyzed the Bio-ID experiment. B.v.E. designed and supervised the project.

## Additional information

**Competing interests:** The authors declare no competing interests.

