## [Peer Review File · Nature Communications]

Reviewers' comments:

Reviewer #1 (Remarks to the Author):

The authors investigated Hippo-independent regulators of Yap activity using a genome-wide CRISPR screen and multiple other genomic approaches. They identified the transcriptional repressor protein Trichorhinophalangeal Syndrome 1 (TRPS1) as a repressor of YAP-dependent transcription. The authors use RNA-Seq and ChIP-Seq and find that TRPS1 globally regulates YAP dependent transcription by binding to a large set of joint genomic sites in enhancer regions. Using mass spectrometry, the authors show that TRPS1 represses YAP-dependent enhancers function by recruiting co-repressor complexes to relevant target genes. They show that TRPS1 inactivation leads to increased enhancer activity with more H3K27 acetylation. Promoter-enhancer interactions, as determined by 4C-seq, are also enhanced with TRPS1 inactivation.

Comments:

1) The authors state that they used the shortest doxycycline induction time in order to limit it to direct YAP targets. This should be further validated by comparing their ChIP seq data to the published YAP ChIP seq data to show the correlation.

2) The authors conclude that TRPS1 represses "the expression of a large set of YAP target genes in breast cancer cells, but a few target genes are affected to a lesser extent or are even spared from this repressive effect."

This is an important point that should be brought out more in the analysis. Can the authors better define the subclasses of Yap targets that they uncover in this study using the data they have already generated?

3) The authors also found that YAP 5SA induction led to a strong recruitment of exogenous YAP to the CTGF promoter. Do the authors see enrichment of Tead elements in the ATAC data? Please present a motif analysis to address this question.

4) The authors performed exogenous co-immunoprecipitation experiments between TRPS1 and the four Gal4-tagged TEAD transcription factors TEAD1-4 (data from Fig. 4d). This should also be validated using endogenous levels of factors.

5) The authors suggest that the ability of TRPS1 to repress YAP/TEAD-dependent target sites is dictated in part by the specific TEAD isoform. In other words certain Teads work better than others to work with TRPS1 to repress Yap. Since most of the data are in overexpression assays this conclusion is not well supported and should be supplemented with more experiments or the statement removed.

Reviewer #2 (Remarks to the Author):

In this manuscript, Elster and colleagues aim to identify novel regulators of YAP signalling independent from the classical Hippo phosphorylation cascade. To do so, they set up an elegant reporter construct in which all the phosphorylation targets have been removed and use this in combination with a genome-wide CRISPR KO experiments. They then investigate mechanistically the first hit from this screen, namely TRPS1, a GATA family repressor. Using a series of well thought biochemical experiments. They show that TRPS1 recruitment at YAP targeted promoters can suppress transcription potentially through epigenetic changes that include nucleosome repositioning and de-acetylation of nucleosomes carrying the active K27ac mark. Finally, they provide some evidence for the functional consequences of TRPS1 removal in terms of 3D chromatin structures. The last part of the manuscript, which is in my opinion the weakest, tries to link genetic aberration at the TRPS1 locus to potential phenotypes in the context of breast cancerogenesis. I consider this work potentially interesting to the Nature Communications

readership once the authors will have addressed some of the key problems with the current versions of the manuscript. Generally, my opinion is that some of the results have been over-interpreted and some of the claims are not well supported by the data.

More specifically, when linking TRSPS1 genomic aberration to breast cancer several essential improvements should be done to the current analysis, otherwise the conclusion should be strongly toned down or removed altogether (as the last figure does not really add to the mechanistic study per se but, currently, appears to be a highly speculative attempt to link function to biology).

1- The authors correctly stated, in the discussion, that TRSPS1 sits in close proximity to Myc and thus they could be co-amplified. If their claim about the prognostic significance and potential role of TRSPS1 is done, then all their analysis should stratify for concurrent Myc amplification and show independent contribution to prognosis. Additionally, they should show that amplification and increased mRNA expression do correlate.

2- Similarly, it would be informative to stratify patients into ER positive and ER negative and run multivariate analysis for subtypes, age, nodal status and other commonly annotated clinical factors now included in TCGA.

3- Provide actual evidence for immune infiltration, from human and mouse model as the current deconvolution assay is not informative as the sampling might have a very strong effect on the results. Since they have the possibility, it would be very informative to test what happens in the 4T1 model with and without TRSPS1.

Additionally, there are some concerns regarding the biochemical part of the study itself. Specifically

1- The authors provide compelling data suggesting that TRSPS1 can lead to chromatin repression by nucleosome remodelling and de-acetylation. Is therefore very counteractive that TRSPS1 overlaps with H3K27ac enhancers to such a strong degree (Fig. 3B). How to the authors explain this?

2- Could the author show what happens to H3K27ac in the absence of TRSPS1(fig 6D) in an alternative manner? As the authors shows only the results at TRSPS1 binding, it would be useful to know if there are MORE H3K27ac de novo sites.

3- Why the authors chose the ANKRD1 promoter for mechanistic studies? Although generating a reporter from this promoter is easier and generally fine for the specific conclusions, it is harder to generalize this finding considering that the author's data strongly suggest TRSPS1 binding is mostly confined to enhancers.

4- It would be useful to link RNA-seq and CHIP-seq and other data as they have the clear target genes.

5- Figure 6h is not informative, as it's just a snapshot and someone cannot evince anything from, say, the H3K27ac panel.

6- Why to the authors keep switching target genes? It would have been good to show what happens to the ANKRD1 in a 4C experiment with the viewpoint in the promoter. Alternatively, what is happening in terms of H3K27ac and other measurement to the IGFBP3 gene?

Minor comments

-Size and scales in many figures are missing (7a and 3i for example).

Overall, I'd be happy to see a revised manuscript that tightens the results presented in Fig. 1-6. As I said, I don't really see the benefit of figure 7 as the data are premature and the results over-stretched and over-interpreted. This section would require a much larger set of experiments.

Reviewer #3 (Remarks to the Author):

In this study, through CRISP screening using MCF10A cells expressing inducible Yap, the authors identified a potential negative regulator of Yap/TEAD-dependent transcription, TRSP1. They further demonstrate that TRSP1 antagonizes Yap-dependent transcription by interaction with TEAD and recruits repressor complex including HDACs. They further show TRSP1 is amplified in some of the breast tumors which is negatively correlated with the Yap gene signature. They also show that TRSP1 is prognostic in breast cancer and patients whose tumors express high levels of TRSP1 turn to have worse survival. The data is interesting as it identifies a novel negative regulator of Yap/TEAD-dependent transcription which seems to have a tumor suppressive activity in breast cancer. However, it remains unclear how the current data help to advance the current understanding whether or not Yap-TEAD truly acts as a tumor suppressor in breast cancer and how exactly is regulated.

Major concerns

Given the tumor heterogeneity of breast cancer and many subtypes of breast cancer, the study was not carefully designed to address the TRSP1-Yap relationship in the context of different subtypes. Is TRSP1 deregulated in all subtypes of breast cancer and its negative relationship applies to all subtypes ?

Loss of YAP activity has been previously shown in MYC high triple negative breast cancer, and this is associated with increased cytoplasmic retention of Yap. In this study, the authors show TRSP1 interaction with TEAD to suppress its transcriptional activity. In what subtype does this mechanism operate ? TNBC or luminal ? Of notice, the authors used MCF-7 and T47D and later using basal mouse 4T1 cells for functional study.

The functional study of TRSP1 is weak and fails to show its dependency on YAP/TEAD. It is unclear why mouse 4T1 is used for the functional study. Is TRSP1 high in 4T1 cells ? Why not using breast cancer cells with high level of TRSP1 ?

Does TRSP1 knockdown-induced growth inhibition is due to Yap-tumor suppressive activity ? Co-knockdown of Yap seems to be necessary to make the point.

Reply to Reviewers

We would like to thank all reviewers for the careful and constructive reviews of our manuscript. In reply to the individual comments, we have included the changes detailed below.

Reviewer 1

1) The authors state that they used the shortest doxycycline induction time in order to limit it to direct YAP targets. This should be further validated by comparing their ChIP seq data to the published YAP ChIP seq data to show the correlation.

As we demonstrate here (like others have done before), YAP and TEAD factors bind mainly to enhancers which complicates A) the direct correlation between ChIP-Seq and RNA-Seq data and B) the comparison with other previously published ChIP-Seq data since every cell line has a very specific set of active enhancers. Unfortunately, there are no YAP ChIP-Seq data in MCF7 cells publically available so that we cannot compare it to publically available data. However, to strengthen this statement, we now included an analysis that tests for the presence of a YAP peak in a 50 kb promoter window of a YAP 5SA-induced gene (Supplementary Fig. 2c). This window is commonly used to capture enhancers controlling a proximal gene. This analysis revealed that ~75% of the YAP-induced genes indeed contain a YAP peak in a 50 kb window surrounding the promoter. Thus, we would reason that our induction time is suitable to largely limit it to direct YAP target genes.

2) The authors conclude that TRPS1 represses “the expression of a large set of YAP target genes in breast cancer cells, but a few target genes are affected to a lesser extent or are even spared from this repressive effect.” This an important point that should be brought out more in the analysis. Can the authors better define the subclasses of Yap targets that they uncover in this study using the data they have already generated?

We now added an MA plot (Fig. 2e) for the RNA-Seq experiments after TRPS1 depletion in MCF7 cells where we highlighted several previously described YAP target genes^{1,2}. Here, it becomes apparent that some target genes are up to 10-fold up-regulated (*ANKRD1*, *IGFBP3*, *NT5E*, *TGFB2*), whereas other well-established YAP target genes are not affected (*CYR61*, *AMOTL2*). Nevertheless, TRPS1 depletion leads to a significant ($P=1.5 \times 10^{-14}$) induction of the YAP target gene set that we defined (now included as a violin plot as Fig. 2f). However, when we performed functional analysis (e.g. GO term analysis) of the YAP target genes (affected vs. unaffected), we were not able to identify functional differences between those target genes. Thus, we did not include this analysis in the current paper.

3) *The authors also found that YAP 5SA induction led to a strong recruitment of exogenous YAP to the CTGF promoter. Do the authors see enrichment of Tead elements in the ATAC data? Please present a motif analysis to address this question.*

We now included a motif analysis (Fig. 3g) that demonstrates an enrichment of two different TEAD binding motifs after TRPS1 depletion in our ATAC-Seq experiments.

4) *The authors performed exogenous co-immunoprecipitation experiments between TRPS1 and the four Gal4-tagged TEAD transcription factors TEAD1-4 (data from Fig. 4d). This should also be validated using endogenous levels of factors.*

We performed two experiments to address this point: proximity ligation assays (PLAs) and endogenous co-immunoprecipitation (Co-IP) experiments. With the PLAs we are able to show that TRPS1 and TEAD1 are able to interact *in situ* (Fig. 4d). Importantly, the PLA signal drops after TRPS1 depletion by siRNA validating the specificity of the assay. Furthermore, we included an endogenous Co-IP between TRPS1 and TEAD proteins (Fig. 4g). These experiments now demonstrate an endogenous interaction between TRPS1 and TEAD proteins after TRPS1 IP in MCF7 cells that is abolished in TRPS1 knockout cells.

5) *The authors suggest that the ability of TRPS1 to repress YAP/TEAD-dependent target sites is dictated in part by the specific TEAD isoform. In other words certain Teads work better than others to work with TRPS1 to repress Yap. Since most of the data are in overexpression assays this conclusion is not well supported and should be supplemented with more experiments or the statement removed.*

We now removed the statement and simply state that TRPS1 gets recruited to joint TRPS1/TEAD sites, probably in a cooperative manner due to a TRPS1-TEAD protein-protein interaction which is further stabilized by additional protein-DNA interactions (TRPS1::GATA; TEAD::TBS)

Reviewer 2

General comment:

Generally, my opinion is that some of the results have been over-interpreted and some of the claims are not well supported by the data.

We now went carefully through the manuscript and toned down several statements/claims to address this point, especially for Fig. 7. Here, we also added several analyses that support our initial hypothesis that TRPS1 affects immunosurveillance.

Specific points:

1- The authors correctly stated, in the discussion, that TRSPS1 sits in close proximity to Myc and thus they could be co-amplified. If their claim about the prognostic significance and potential role of TRSPS1 is done, then all their analysis should stratify for concurrent Myc amplification and show independent contribution to prognosis. Additionally, they should show that amplification and increased mRNA expression do correlate.

When we reanalyzed the TCGA data sets for concurrent Myc amplification, we were not able to stratify the patients according to the reviewer's suggestion since *MYC* and *TRPS1* seem to be nearly always co-amplified (see Supplementary Fig. 7b,c). Even though this complicates our analyses, we believe that the tumour might select for this large amplicon. Otherwise, one would expect to find focal Myc amplifications, which does not seem to be the case. It has previously been shown in mouse models that *Myc* amplification alone does not accelerate *MMTVneu*-driven breast tumour progression but depends on additional co-amplified genes such as *Pvt1*³. Consequently, TRPS1 might be an additional player of this amplicon, even though we do not have formal evidence for this hypothesis. We now added this in the discussion section. Additionally, we added an analysis that demonstrates a correlation between TRPS1 amplification and increased TRPS1 mRNA expression (Fig. 7c).

2- Similarly, it would be informative to stratify patients into ER positive and ER negative and run multivariate analysis for subtypes, age, nodal status and other commonly annotated clinical factors now included in TCGA.

We now included analyses in which we stratified according to ER status or nodal status (Supplementary Fig. 7d,e). Here, we could not identify any significant changes after stratification, indicating that TRPS1 is an independent prognostic factor (now also included also in the text).

3- Provide actual evidence for immune infiltration, from human and mouse model as the current deconvolution assay is not informative as the sampling might have a very strong effect on the results. Since they have the possibility, it would be very informative to test what happens in the 4T1 model with and without TRSPS1.

We analysed TRPS1 expression in 17 additional human breast tumours by immunohistochemistry. Consistent with our previous statement, the pathologist Prof. Andreas Rosenwald could verify high TRPS1 expression in all 27 breast cancer patients analysed. However, according to his statement, he was not able to stratify TRPS1^{high} vs. TRPS1^{low} patients, probably because the IHC analyses are not quantitative enough to detect the differences that we see in the TCGA data set using bioinformatic approaches. Thus, we performed IHC analyses of the 4T1 tumours. Here, we focused on CD3 because it marks both CD4 and CD8 cells whose signatures are both negatively correlated with TRPS1 activity in our deconvolution analyses of the TCGA RNA-Seq data set in Fig. 7m (to which we now also added the CD4 signature).

Our experiments demonstrate that TRPS1 depletion significantly increased the frequency of intratumoural CD3-positive cells, now included as Fig. 7k,l. Thus, these analyses provide evidence that TRPS1 can impact on tumour immunosurveillance.

Concerns regarding the biochemical analyses

1- The authors provide compelling data suggesting that TRSPS1 can lead to chromatin repression by nucleosome remodelling and de-acetylation. Is therefore very counteractive that TRSPS1 overlaps with H3K27ac enhancers to such a strong degree (Fig. 3B). How to the authors explain this?

Even though it seems counterintuitive, similar observations have been made for HDACs at promoters since HDACs are more abundant at the promoters of heavily transcribed genes compared to weakly transcribed genes⁴. Thus, TRPS1 is not an exception here. We believe that TRPS1 binds to a specific set of active enhancers to restrain their activity (as demonstrated by our 4C-Seq analyses) and does not shut them down completely.

2- Could the author show what happens to H3K27ac in the absence of TRSPS1(fig 6D) in an alternative manner? As the authors shows only the results at TRSPS1 binding, it would be useful to know if there are MORE H3K27ac de novo sites.

We now included an analysis for H3K27ac peaks in MCF WT cells compared to TRPS1 KO cells (Fig. 6h). Here, deletion of TRPS1 leads to additional 15,297 H3K27ac peaks in TRPS1

KO cells compared to 5,971 additional H3K27ac peaks in MCF7 WT cells. Even though secondary effects might also contribute to these changes (e.g. they could also contribute to the additional 5,971 peaks seen in MCF7 WT cells), this suggests that TRPS1 indeed restrains H3K27ac at specific sites.

3- Why the authors chose the ANKRD1 promoter for mechanistic studies? Although generating a reporter from this promoter is easier and generally fine for the specific conclusions, it is harder to generalize this finding considering that the author's data strongly suggest TRPS1 binding is mostly confined to enhancers.

We now substituted the whole ANKRD1 reporter analysis (and moved it to Supplementary Fig. 4) for reporter analyses using an enhancer fragment. Essentially, we were able to corroborate our previous findings that TRPS1 is able to repress YAP/TEAD-dependent transcription, also at enhancers.

4- It would be useful to link RNA-seq and CHIP-seq and other data as they have the clear target genes.

We now included an analysis combining our RNA-Seq with our ChIP-Seq experiments to address this point (Supplementary Fig. 3g). Here, we defined two sets of TRPS1 target genes: genes that have a TRPS1 peak within a ± 50 kb window (including at least a fraction of TRPS1 target genes driven by a distant enhancer) or a ± 1 kb window (largely consisting of TRPS1-bound promoters). As control group, we used genes that do not have a TRPS1 peak in a ± 50 kb window. Here, we can demonstrate that depletion of TRPS1 leads to increased expression of TRPS1 target genes (± 50 kb and ± 1 kb set) compared to the control group. Interestingly, the ± 50 kb TRPS1 target genes were more strongly affected than the ± 1 kb target gene set, suggesting that TRPS1-mediated effects on enhancers are more pronounced than TRPS1-dependent repression at promoters.

5- Figure 6h is not informative, as it's just a snapshot and someone cannot evince anything from, say, the H3K27ac panel.

We now removed this panel.

6- Why to the authors keep switching target genes? It would have been good to show what happens to the ANKRD1 in a 4C experiment with the viewpoint in the promoter. Alternatively, what is happening in terms of H3K27ac and other measurement to the IGFBP3 gene?

We performed 4C-Seq for the *ANKRD1* promoter. However, *ANKRD1* has a promoter peak for YAP/TEAD and TRPS1 (see Supplementary Fig. 5a) and one would not expect TRPS1 deletion to have a strong effect on long-range interactions. Furthermore, we could not detect any long-range interactions using the *ANKRD1* promoter as bait (see Figure for Reviewer 1). This also becomes evident in our 4C analysis since we sequenced many fragends that were derived from a “self-ligated” construct – an indicator that the viewpoint does not interact with distant sites but rather generates self-ligated products. Thus, we did not include *ANKRD1* 4C-Seq experiments.

When we analyzed the enhancers that lie in the region of the TAD, we could identify several enhancers that demonstrated an increased H3K27ac signal in the TRPS1 KO cells. However, it is hard to tell which enhancer is the one that is responsible for the effect in our 4C experiment. It could also be a result of a combinatorial effect of several affected enhancers. To make that point clear to the reader, we now added all the enhancers and TRPS1 peaks that lie in the region (in Fig. 6 k as pink and blue bars, respectively). We also state in the text that “*We were not able to correlate this effect with altered activity of a single enhancer, probably due to the fact that dozens of enhancers lie in the genomic region of the IGFBP3 locus*”.

By removing the *ANKRD1* luciferase assays from the main figures (now in Supplementary Fig. 4) and including a functional reporter analysis of an enhancer bound by TRPS1 and YAP/TEAD1 (Fig. 6i,j), we also reduced the problem of target gene switching.

-Size and scales in many figures are missing (7a and 3i for example).

We added scale bars to the respective figures.

Reviewer 3

1.)

Given the tumor heterogeneity of breast cancer and many subtypes of breast cancer, the study was not carefully designed to address the TRSP1-Yap relationship in the context of different subtypes. Is TRSP1 deregulated in all subtypes of breast cancer and its negative relationship applies to all subtypes?

We now included an additional analysis where we stratified the patients according to their breast cancer subtype: Basal, Her2, LumA, LumB and Normal (Fig. 7e). We are able to demonstrate with the help of these analyses that the negative relationship between TRPS1 and YAP1 applies to all subtypes. Nevertheless, the effect of TRPS1 on its target genes seems to be more pronounced in LumB and Basal subtypes since TRPS1 target genes are more strongly repressed in those subtypes compared to the other subtypes (now included in Fig. 7f). From a clinical point of view, this is very interesting since basal tumours are known to be very aggressive and the Luminal B is much more aggressive than the Luminal A subtype.

2.)

Loss of YAP activity has been previously shown in MYC high triple negative breast cancer, and this is associated with increased cytoplasmic retention of Yap. In this study, the authors show TRSP1 interaction with TEAD to suppress its transcriptional activity. In what subtype does this mechanism operate ? TNBC or luminal ? Of notice, the authors used MCF-7 and T47D and later using basal mouse 4T1 cells for functional study.

As described above, the inhibitory effect of TRPS1 on YAP/TEAD-dependent transcription does not seem to be restricted to any specific breast cancer cell type. Of note, the CRISPR screen and the initial validation (Fig. 1h) were performed in MCF10A cells, a primary basal breast epithelial cell line. Thus, we do not believe that TRPS1 only operates in a specific subtype but rather that cancer cells hijack this program to restrain YAP/TEAD activity.

Our initial analysis revealed high TRPS1 expression on protein level in MCF7 and T47D cells. That is why we performed most of our mechanistic studies in these two cell lines. As we pointed out, we also see repression of YAP target genes by TRPS1 in MCF10A cells. That is why we think this is a general mechanism.

For the *in vivo* analyses we used 4T1 cells for two reasons: First, from our previous analyses (now included in Fig. 7f) we knew that TRPS1 is highly active in basal tumours. Second, 4T1 cells were previously used by the Guan lab to demonstrate an effect of YAP on immunosurveillance⁵. Thus - in conjunction with the clinical data, demonstrating high

TRPS1 activity in basal tumours - we consider the 4T1 model a suitable model to test that hypothesis TRPS1 depletion recapitulates aspects of high YAP activity.

3.)

The functional study of TRSP1 is weak and fails to show its dependency on YAP/TEAD. It is unclear why mouse 4T1 is used for the functional study. Is TRSP1 high in 4T1 cells ?

For an explanation why we used the 4T1 model, see 2.).

4.)

Why not using breast cancer cells with high level of TRSP1?

The Guan lab showed that even though high YAP activity leads to increased tumorigenic potential in *in vitro* transformation assays, YAP displayed tumour suppressive functions when injected into immunocompetent mice using syngeneic mouse models ⁵. This was linked to YAP's effect on the immune system.

Based on this, we had to use a model that allows us to inject tumour cells into immunocompetent mice. Consequently, we could not use human breast cancer cells, which would need to be injected into immunocompromised mice, e.g. nude mice.

5.)

Does TRSP1 knockdown-induced growth inhibition is due to Yap-tumor suppressive activity? Co-knockdown of Yap seems to be necessary to make the point.

We cloned a panel of tandem shYAP/TAZ constructs, similar to the strategy that we successfully used in our previous Cancer Cell paper, to co-deplete YAP and TAZ ⁶. However, when we infected 4T1 cells with this construct, we only obtained cells that showed a poor knockdown. Strikingly, many cells died during the selection process even though they were infected (based on the expression of GFP which is part of the mirE shRNA strategy). In contrast, we were able to achieve excellent knockdown efficiencies with the same constructs in NIH3T3 cells (one example of the most potent shRNA is given as a figure below as Fig. 2 in the Figures for Reviewers section). Thus, it seems that 4T1 cells select against YAP/TAZ depletion, which makes the double depletion an unfeasible experiment. This result implies that 4T1 cells (at least for efficient growth *in vitro*) need a certain degree of YAP/TAZ activity.

We absolutely agree with the reviewer's comment that we cannot conclude from these experiments that the effects observed upon TRPS1 depletion are connected to the effect of TRPS1 depletion on YAP activity. Therefore, we cannot draw this conclusion, which we also mention in the discussion part. However, given the effect of TRPS1 depletion on immune cell infiltration, we believe that it is reasonable to suggest such a hypothesis.

Figures for Reviewers

Figure 1

Fig. 1: 4C-Seq experiments using the ANKRD1 transcriptional start site (TSS) as bait. The experiment was performed in MCF7 WT cells and TRPS1 KO cells.

Figure 2

Fig. 2: Western Blots from 4T1 cells (left) and NIH3T3 cells (right) expressing the indicated tandem shRNAs targeting YAP TAZ.

References

1. Zhao, B. *et al.* TEAD mediates YAP-dependent gene induction and growth control. *Genes Dev.* **22**, 1962–1971 (2008).
2. Mohseni, M. *et al.* A genetic screen identifies an LKB1-MARK signalling axis controlling the Hippo-YAP pathway. *Nat. Cell Biol.* **16**, 108–117 (2014).
3. Tseng, Y.-Y. *et al.* PVT1 dependence in cancer with MYC copy-number increase. *Nature* **512**, 82–86 (2014).
4. Wang, Z. *et al.* Genome-wide mapping of HATs and HDACs reveals distinct functions in active and inactive genes. *Cell* **138**, 1019–1031 (2009).
5. Moroishi, T. *et al.* The Hippo Pathway Kinases LATS1/2 Suppress Cancer Immunity. *Cell* **167**, 1525–1539.e17 (2016).
6. Eyss, von, B. *et al.* A MYC-Driven Change in Mitochondrial Dynamics Limits YAP/TAZ Function in Mammary Epithelial Cells and Breast Cancer. *Cancer Cell* **28**, 743–757 (2015).

Reviewers' comments:

Reviewer #1, Expertise : Hippo/ YAP pathway (Remarks to the Author):

concerns adequately addressed - very interesting paper that adds to our understanding of Yap mediated transcriptional regulation

Reviewer #2, Expertise: Epigenomics
(Remarks to the Author):

The authors have tried to address more or less successfully most of my comments.

"We now went carefully through the manuscript and toned down several statements/claims to address this point, especially for Fig. 7. Here, we also added several analyses that support our initial hypothesis that TRPS1 affects immunosurveillance"

It would be very helpful if the authors spelled out where these changes occurred

Specific Point 1.

Overall, this is the biggest concern I have and I still strongly feel the authors should drop this point altogether. They have no data to support the conclusions. Their argument that as the amplicon is bigger than MYC alone suggest that TRPS1 is important is flawed (it could be any of the other genes, for example SQLE has been shown to have independent prognostic value even when correcting for MYC). At the end of the day, they candidly state that they "do not have formal evidence for this hypothesis". Hence, they can discuss the speculation but I think they should remove the paragraph and data has the add nothing to the paper.

Specific point 2- Supplementary figure 7d and e are not multivariate analysis, they should run KM analysis correcting for the clinical variables (what about grade? age? Size?). I think this is essential before the paper is accepted.

Specific point 3- I am not an immunologist, hence I cannot comment on the results. I am pleased the authors have run the experiment but the results are not impressive (at least to me) as it's an n=1 (1 mouse model) with only 4 sections stained, and the results indicate that CD3 positive cells go from an average of 2 per section to 6 per section. Is this really a strong signal that the immune system has now flooded the cancer? As I say, an immunologist can comment with more authority than me, but I can say that the effect seems modest.

Reviewer #3, Expertise: Breast cancer (Remarks to the Author):

the revised manuscript has addressed some of my questions but I am still not convinced of the functional part

It remains unclear why the authors chose MCF7 and T47D for functional studies. Myc is not really amplified in these two cell lines and if TRPS1 is co-amplified with Myc we will expect to see the amplification of TRPS1 in MCF7 and T47D but the author never tried to specify it. There are other well-known MYC amplified breast cancer cell lines that the authors should examine to show the TRPS1 dependency. Ideally, I would choose a few Myc dependent breast cell lines to make the point.

4T1 in vitro data is also missing

Again the functional study is performed superficially.

Response to Reviewers' comments:

We would like to thank the reviewers again for their work and their constructive suggestions. Please find our response to the individual points below.

Reviewer #1

concerns adequately addressed - very interesting paper that adds to our understanding of Yap Tead transcriptional regulation

Thank you very much.

Reviewer #2

"We now went carefully through the manuscript and toned down several statements/claims to address this point, especially for Fig. 7. Here, we also added several analyses that support our initial hypothesis that TRPS1 affects immunosurveillance"

It would be very helpful if the authors spelled out where these changes occurred

We performed the following changes in the revised manuscript:

Rephrased:

Originally:

Thus, we hypothesized that the oncogenic function of TRPS1 results from its repressive function on YAP target genes and the associated effects on immunosurveillance.

Rephrased:

Thus, we hypothesized that TRPS1's oncogenic role could at least in part result from its repressive function on the transcriptional output of YAP.

Statements removed:

- It is nevertheless tempting to speculate that the lineage commitment defects observed in the mesenchymal cells of TRPS1 patients/TRPS1-deficient mice might also arise from deregulated YAP/TAZ activity, since YAP and TAZ are key regulators of mesenchymal stem cell differentiation.

Additions to Fig. 7:

- Thus, we analysed the expression pattern of TRPS1 in sections of breast cancer samples and adjacent normal tissue (Fig. 7a; Supplementary Fig. 7a). Here, TRPS1 expression was confined to the luminal compartment in normal tissue. TRPS1 demonstrated a very strong nuclear staining in 27 out of 27 breast cancer patients, indicating that TRPS1 expression does not commonly get lost in breast cancer.

- This is supported by our analyses of the publicly available TCGA data sets for breast cancer which show that the TRPS1 locus is not deleted in human breast cancer patients but is rather amplified in a high fraction of patients leading to increased TRPS1 mRNA expression (Fig. 7b,c).

- The anti-correlation between TRPS1 and YAP activity was neither confined to a specific breast cancer subtype (Fig. 7e).

- Strikingly, TRPS1 target genes were most strongly repressed in the Luminal B and Basal subtype (Fig. 7f), two rather aggressive subtypes.

- TRPS1 depletion was associated with a significantly increased frequency of intra-tumoural T cells as determined by immunohistochemical stainings for CD3 (Fig. 7k,l).

- Importantly, gene expression deconvolution of TCGA RNA-Sequencing also revealed a relationship between high TRPS1 activity and low frequency of intra-tumoural immune cells (Fig. 7m) since tumours demonstrating high TRPS1 activity, showed a strongly reduced frequency of CD4, CD8 and natural killer (NK) cells, respectively (Fig. 7m).

Changes in the discussion:

Originally:

Although it is conceivable that tumour cells down-regulate YAP's activity to escape the immune surveillance mechanisms, it is hard to reconcile this with YAP's and TAZ's ability to confer cancer stem cell traits to breast epithelial cells, e.g. induction of sphere forming capacity. We propose that a small fraction of cells within tumours are still able to maintain high YAP/TAZ activity. This fraction needs to be small enough to not induce an immune response but big enough to maintain the tumour. In the future it will be important to identify these cells by suitable reporter approaches in a complex mixture of tumour cells and to analyse how their TRPS1 and MYC activity might be restrained.

Changed to:

Although it is conceivable that tumour cells down-regulate YAP activity to escape the immunosurveillance mechanisms, it is hard to reconcile this with YAP/TAZ ability to confer cancer stem cell traits to breast epithelial cells and its pro-oncogenic functions. We propose that breast cancer cells need to maintain a certain level of YAP/TAZ activity that is high enough to maintain the pro-oncogenic functions of YAP/TAZ but sufficiently low to escape immunosurveillance. Thus, TRPS1 could exert its oncogenic role by keeping YAP/TAZ activity in check.

Sentence added to make clear that we believe that TRPS1 most certainly has YAP/TAZ-independent functions, probably also in breast cancer:

In the future, it will be important to discriminate between YAP/TAZ-dependent and YAP/TAZ-independent functions for the oncogenic role of TRPS1 in vivo.

Specific Point 1.

Overall, this is the biggest concern I have and I still strongly feel the authors should drop this point altogether. They have no data to support the conclusions. Their argument that as the amplicon is bigger than MYC alone suggest that TRPS1 is important is flawed (it could be any of the other genes, for example SQLE has been shown to have independent prognostic value even when correcting for MYC). At the end of the day, they candidly state that they "do not have formal evidence for this hypothesis". Hence, they can discuss the speculation but I think they should remove the paragraph and data has the add nothing to the paper.

We included this point due to an initial request by reviewer #3. Since according to reviewer #2 this point is distracting and adds nothing to the story, we now removed this part of the manuscript (Supp. Fig. 7b,c). However, we also agree with reviewer #3 that the location of the *TRPS1* locus is important with respect to the amplicon harbouring the *MYC* gene. We believe that we should make the reader aware of this and we consequently still mention this fact in the discussion part.

Specific point 2- Supplementary figure 7d and e are not multivariate analysis, they should run KM analysis correcting for the clinical variables (what about grade? age? Size?). I think this is essential before the paper is accepted.

To address this point, we performed multivariate analysis on a publically available breast cancer patients data (<http://co.bmc.lu.se/gobo/gobo.pl>) set comprising 1,881 breast cancer patients¹.

This analysis demonstrates that high TRPS1 activity (based on low expression of TRPS1-repressed target genes) is correlated with a decreased recurrence-free survival of breast cancer patients (now in Supplementary Fig. 7b). Importantly, the multivariate analysis revealed that high TRPS1 activity is an independent predictor of survival when we analysed all patients (Supplementary Fig. 7c). The same was true for an identical analysis performed with lymph node-positive patients (Supplementary Fig. 7d,e). Due to space constraints, we could not include the complete analysis as a full figure. Therefore, we included a summary of the multivariate analysis as a supplementary table (Supplementary Table 1).

Specific point 3- I am not an immunologist, hence I cannot comment on the results. I am pleased the authors have run the experiment but the results are not impressive (at least to me) as it's an n=1 (1 mouse model) with only 4 section stained, and the results indicate that CD3 positive cells go from an average of 2 per section to 6 per section. Is this really a strong signal that the immune system has now flooded the cancer? As I say, an immunologist can comment with more authority than me, but I can say that the effect seem modest.

The effect on CD3-positive cells might not seem very strong but this effect size has also recently been reported in a study that investigated the effect of PD-L1 glycosylation on tumour-infiltrating lymphocytes using the 4T1 model². In Fig. 7H of this paper, the authors demonstrate an approx. 2-fold increase in a subset of CD3-positive of T cells leading to a decreased primary tumour growth *in vivo*, similar to what we describe in Fig. 7h-j of our manuscript (Fig. 7G in Li et al.).

As we also pointed out in our response to reviewer #3, we are aware of the fact that the *in vivo* work is done in one mouse model, even though this model is widely used in breast cancer research. In the future, it will be important to investigate the oncogenic role of TRPS1 using different mouse models, e.g. by the use of an inducible TRPS1 knockout/knockdown model in combination with different oncogenic driver mutations and down-regulating TRPS1 during different times of tumour development.

Reviewer #3 (Remarks to the Author):

the revised manuscript has addressed some of my questions but I am still not convinced of the functional part

It remains unclear why the authors chose MCF7 and T47D for functional studies. Myc is not really amplified in these two cell lines and if TRPS1 is co-amplified with Myc we will expect to see the amplification of TRPS1 in MCF7 and T47D but the author never tried to specify it.

There are other well-known MYC amplified breast cancer cell lines that the authors should examine to show the TRPS1 dependency. Ideally, I would choose a few Myc dependent breast cell lines to make the point.

We picked these two cell lines since - according to the cell line database of the Sanger Institute (https://cancer.sanger.ac.uk/cell_lines/) - A) these two cell lines demonstrate an up-regulation of *TRPS1* mRNA expression (see Figure 1 for Reviewer), plus B) we can demonstrate high TRPS1 expression on protein level (Supp. Fig. 2a).

According to the Sanger Institute data, *TRPS1* mRNA is up-regulated in MCF7 and T47D cells despite the absence of a detectable *TRPS1* amplification. This suggests that there could exist additional mechanisms for tumour cells to up-regulate *TRPS1* expression and that TRPS1 amplification is a first hint for a potentially oncogenic role of this factor. However, it seems that amplification is not the only mechanism leading to high TRPS1 expression.

As the reviewer correctly pointed out, T47D and MCF7 cells carry no *MYC* amplification and show no *MYC* overexpression, which in our view strengthens our point that tumour cells/ tumour cell lines actively select for high TRPS1 expression and that increased TRPS1 expression is not solely a “bystander” effect of *MYC* amplification.

We therefore believe that we would not gain any additional information regarding TRPS1's oncogenic role by using MYC-dependent breast cancer cell lines since: A) the scope of this manuscript is not MYC's role or the 8q amplicon and B) we can see increased TRPS1 expression independent of the 8q amplicon in the two cell lines that we used in the biochemical characterization.

Dissecting the specific contribution of every single gene in this amplicon would require a proper genetic mouse model as it was previously shown for the Pvt1/Myc connection³ and would be (in our opinion) beyond the scope of this manuscript.

4T1 in vitro data is also missing

We are not sure what is meant here? We now included a qRT-PCR analysis that compares TRPS1 expression in FACS-purified mouse mammary epithelial cells (MECs) with TRPS1 expression in 4T1 cells (Supplementary Fig. 7f). This analysis revealed an approx. 4-fold higher TRPS1 expression in 4T1 tumour cells compared to mouse MECs pointing towards a potentially oncogenic role of TRPS1 in 4T1 cells, which we can confirm by the orthotopic injection of TRPS1-depleted 4T1 cells.

Again the functional study is performed superficially.

We agree, that the oncogenic role of TRPS1 needs further in depth *in vivo* analysis, e.g. by the use of a genetic *Trps1* mouse model. However, by including the 4T1 data our intention is to provide first evidence for an oncogenic role of TRPS1. As our biochemical data demonstrate that TRPS1 is able to antagonize YAP/TEAD-dependent transcription, we chose the 4T1 model because A) this model was used to demonstrate a tumour-suppressive role of YAP⁴ and B) we had to use an immunocompetent breast cancer model since our bioinformatics analyses of TCGA data demonstrated a correlation between TRPS1 activity, YAP activity and infiltrating immune cells (Fig. 7e-m).

Figure 1 for reviewer:

MCF7 MYC

https://cancer.sanger.ac.uk/cell_lines/sample/overview?id=905946

MCF7 TRPS1

T47D MYC

https://cancer.sanger.ac.uk/cell_lines/sample/overview?id=905945

T47D TRPS1

Figure 1:

Snapshots of the Gene browser section of the COSMIC (Catalogue of Somatic Mutations In Cancer) database showing copy number variation (CNV gain) and potential overexpression (gene expression - over) for *TRPS1* and *MYC* in the breast cancer cell lines MCF7 and T47D, respectively.

References

1. Ringnér, M., Fredlund, E., Häkkinen, J., Borg, Å. & Staaf, J. GOBO: gene expression-based outcome for breast cancer online. *PLoS ONE* **6**, e17911 (2011).
2. Li, C.-W. *et al.* Eradication of Triple-Negative Breast Cancer Cells by Targeting Glycosylated PD-L1. *Cancer Cell* **33**, 187–201.e10 (2018).
3. Tseng, Y.-Y. *et al.* PVT1 dependence in cancer with MYC copy-number increase. *Nature* **512**, 82–86 (2014).
4. Moroishi, T. *et al.* The Hippo Pathway Kinases LATS1/2 Suppress Cancer Immunity. *Cell* **167**, 1525–1539.e17 (2016).

REVIEWERS' COMMENTS:

Reviewer #2 (Remarks to the Author):

I am satisfied by the authors reviews with the condition that they remove the MYC subplot (Specific point 1).

Response to Reviewers' comments:

Reviewer #2

"I am satisfied by the authors reviews with the condition that they remove the MYC subplot (Specific point 1)"

We removed all Myc data and Myc subplots from the Results section.